# Inhibiting heme piracy by pathogenic *Escherichia coli* using de novo-designed proteins

Daniel R. Fox [1,2,3], Kazem Asadollahi[3], Imogen Samuels[3], Bradley A. Spicer [4], Ashleigh Kropp[1,2], Christopher J. Lupton [4], Kevin Lim[5], Chunxiao Wang[3], Hari Venugopal [6], Marija Dramicanin[5,7], Gavin J. Knott [4] ✉ & Rhys Grinter [1,2,3] ✉

Iron is an essential nutrient for most bacteria and is often growth-limiting during infection, due to the host sequestering free iron as part of the innate immune response. To obtain the iron required for growth, many bacterial pathogens encode transporters capable of extracting the iron-containing cofactor heme directly from host proteins. Pathogenic *E. coli* and *Shigella spp.* produce the outer membrane transporter ChuA, which binds host hemoglobin and extracts its heme cofactor, before importing heme into the cell. Heme extraction by ChuA is a dynamic process, with the transporter capable of rapidly extracting heme from hemoglobin in the absence of an external energy source, without forming a stable ChuA-hemoglobin complex. In this work, we utilise a combination of structural modelling, Cryo-EM, X-ray crystallography, mutagenesis, and phenotypic analysis to understand the mechanistic detail of this process. Based on this understanding we utilise artificial intelligence-based protein design to create binders capable of inhibiting *E. coli* growth by blocking hemoglobin binding to ChuA. By screening a limited number of these designs, we identify several binders that inhibit *E. coli* growth at low nanomolar concentrations, without experimental optimisation. We determine the structure of a subset of these binders, alone and in complex with ChuA, demonstrating that they closely match the computational design. This work demonstrates the utility of de novo-designed proteins for inhibiting bacterial nutrient uptake and uses a workflow that could be applied to integral membrane proteins in other organisms.

*Escherichia coli* and the closely related genus *Shigella* (containing the species *S. boydii, S. dysenteriae, S. flexneri,* and *S. sonnei*) are important pathogens of humans and animals[1–4]. *E. coli* and *Shigella* are genetically diverse with different strains sharing as little as 40% of their protein-encoding genes[2,5]. This diversity allows these bacteria to adopt a range of lifestyles from harmless commensal to intestinal and extra-intestinal pathogen[2,6]. *E. coli* strains that cause intestinal infections fall into six

subtypes (DAEC, EAEC, EHEC, EIEC, EPEC, ETEC) based on lineage and disease symptoms[2,6]. EIEC is closely related to *Shigella spp.*, which emerged multiple times from *E. coli* ancestors, and have limited genetic differences[7]. Pathogenic *E. coli* and *Shigella* are major causes of diarrheal mortality in children in developing regions and of foodborne infections worldwide[4,8]. In 2016, *Shigella* caused 270 million cases and 212,000 deaths, ETEC caused 222 million cases and 51,000 deaths, and

EPEC 14 million cases and 12,000 deaths[9,10]. Extra-intestinal *E. coli* (ExPEC) is the primary cause of urinary tract infections (UPEC) and neonatal meningitis (NMEC), contributing significantly to disease burden[2,11]. Increasing antimicrobial resistance among pathogenic *E. coli* and *Shigella* is a major concern, with the World Health Organization ranking *E. coli* as third on the list of twelve antibiotic-resistant 'priority pathogens' and designating fluoroquinolone-resistant *Shigella spp.* high priority status in the Bacterial Priority Pathogens List 2024[12,13].

Like most bacterial pathogens, *E. coli* and *Shigella* require the essential nutrient iron to cause infection[14–17]. Most bacteria require iron as a cofactor for enzymes that perform diverse reactions key to their survival and persistence. For example, DNA biogenesis, gene regulation and respiratory ATP production all require enzymatic reactions dependent on iron[18]. While available sources of iron differ depending on the host context, it is a scarce resource due to sequestration by the host, via an innate immune process known as nutritional immunity[19]. *E. coli* and *Shigella* employ two general strategies to obtain iron during infection. In the first strategy, they secrete compounds known as siderophores, which sequester iron with high affinity, and then reimport and process the iron-siderophore complex[20,21]. In the second strategy, they import the iron-containing cofactor heme, either as free heme or by extraction from host heme-containing proteins like hemoglobin[22]. Both these strategies rely on a class of outer-membrane protein known as TonB-dependent transporters (TBDTs)[23,24]. TBDTs bind a target nutrient compound with high affinity and import it into the cell using energy provided by the inner-membrane TonB-ExbBD complex[25]. TBDTs consist of a 22-stranded transmembrane β-barrel with a pore occluded by a globular plug domain, which is transiently displaced during import[23,24]. While the structure of the TBDT barrel is highly conserved, different TBDTs have diverse extracellular loops that mediate a high level of substrate specificity, while the transient displacement of the plug domain prevents the ingress of noxious compounds[23,24]. While *E. coli* and *Shigella* strains encode many genes for the production and import of chemically diverse siderophores[20,21], only two TBDTs that import heme have been identified: Hma which imports free heme, and ChuA (aka. ShuA) that imports either free heme or extracts heme directly from host hemoglobin[14–17,26,27]. Hma and ChuA are important for virulence, as mutants lacking *hma* and *chuA* were outcompeted by the WT parental strain in the kidney during co-infection experiments in mice[27,28].

ChuA interacts transiently with host hemoglobin and rapidly extracts its heme cofactor[16]. It is also capable of extracting heme from myoglobin although at a much slower rate. Two histidine residues (His-86 and His-420) are essential for this process[16]. His-420 is located in extracellular loop 7, while His-86 is located in the plug domain[17]. The presence and location of these histidine residues imply a transition between distinct coordination states occurs during heme extraction and import. However, the structural basis for hemoglobin binding, heme extraction and import remained unresolved. In this work, we utilise a combination of structural modelling, Cryo-EM, X-ray crystallography, mutagenesis, and phenotypic analysis to determine the mechanistic detail of this process.

Preventing bacterial growth by blocking access to essential nutrients remains an underexplored strategy, which could be exploited to develop novel treatments for infections caused by antibiotic-resistant bacterial pathogens. Leveraging our understanding of the molecular basis for ChuA function, we utilised RFdiffusion-based protein design to create binders that block hemoglobin binding to ChuA[29]. We screened a limited number of these designs, identifying several binders that inhibit *E. coli* growth at low nanomolar concentrations when hemoglobin or myoglobin is the sole available iron source. We characterised the affinity of a subset of these binders for ChuA, and determined representative structures, alone and in complex with ChuA, demonstrating that they closely match the computational design. This work demonstrates the utility of de novo-designed proteins for inhibiting the growth of bacterial pathogens by blocking the import of essential nutrients. Moreover, it demonstrates the utility of AI-based protein design to create binders capable of modulating the function of membrane transporters, using a workflow that could be applied to integral membrane proteins in other organisms.

## Results

### ChuA targets hemoglobin as its high-affinity substrate

Previous phenotypic analysis indicates ChuA acts as a heme transporter, targeting either free hemin (ferric heme) or hemoglobin[14–17], while biochemical analysis indicates that ChuA binds free hemin much more slowly than it extracts heme from hemoglobin, suggesting free hemin may not be the transporter's preferred target for extracellular uptake[16]. In addition, ChuA also extracts heme from myoglobin, although at a much slower rate than from hemoglobin, suggesting this protein is also a ChuA substrate[16]. To reconcile these data, we sought to determine the substrate specificity of ChuA in bacterial cell culture. However, redundancy in TBDT-based iron-uptake systems makes it difficult to assess the contribution of a specific transporter to iron uptake. To solve this, we utilised an *E. coli* BW25113 strain that lacks all TBDTs involved in iron uptake (*E. coli$_{ΔTBDT}$*)[30–32]. *E. coli* BW25113 does not naturally possess the Chu operon, which encodes ChuA and other proteins required heme import, so we inserted this operon by homologous recombination[14,33] (Fig. 1a). This strain is impaired in its ability to grow under even mildly iron-limited conditions[30–32], providing a clean background for assessing ChuA function. We tested the ability of this strain (*E. coli$_{ΔTBDT:ChuOP}$*) to grow on agar containing free hemin or various human heme or iron-containing proteins as its iron source. In this assay, *E. coli$_{ΔTBDT:ChuOP}$* was able to grow with hemoglobin (adult αβ or fetal αγ), myoglobin and hemin as a heme/iron source, but not cytoglobin, neuroglobin, or ferredoxin 1 (Fig. 1b). The ability to grow on hemoglobin, myoglobin and hemin was abolished in a *chuA* knockout (*E. coli$_{ΔTBDT:ChuOP:ΔchuA}$*), but restored by complementation with a plasmid-encoded copy of *chuA*, demonstrating the specificity of this effect (Fig. 1b, and Supplementary Fig. 1a).

Next, we assessed the growth of *E. coli$_{ΔTBDT:ChuOP}$* in liquid culture across a range of heme-containing substrate concentrations. While neither the parental *E. coli$_{ΔTBDT}$* strain nor *E. coli$_{ΔTBDT:ChuOP:ΔchuA}$* were able to grow at any substrate concentration, *E. coli$_{ΔTBDT:ChuOP}$* exhibited growth in the presence of hemoglobin (adult αβHb or fetal αγHb), myoglobin (Mb), hemin, and to a lesser extent neuroglobin (Figure 1b, and Supplementary Fig. 1b). The affinity of ChuA for these substrates varied considerably, with αβHb and αγHb representing the highest affinity substrates (EC$_{50}$ values of 53 and 44 nM respectively), followed by hemin (EC$_{50}$ value of 350 nM), and myoglobin (EC$_{50}$ value of 1.3 μM) (Fig. 1b). These data indicate that while ChuA can import heme derived from several sources, hemoglobin represents its highest affinity substrate. To determine the affinity of ChuA for αβHb and Mb, we performed bio-layer interferometry (BLI), with recombinant globins immobilised on the sensor. We measured a disassociation constant ($K_D$) for ChuA binding to αβHb of 71.5 nM, with relatively fast association ($k_{on}$) and dissociation rate ($k_{off}$) constants, consistent with the transient nature of the interaction (Supplementary Fig. 1c, Supplementary Table 1). Some signal was detected for Mb but with insufficient response to measure binding kinetics, indicating that ChuA interacts more weakly with this substrate.

### ChuA binds heme and hemoglobin via extracellular loops

Based on the observation that ChuA has the highest affinity for hemoglobin, we attempted to determine the structure of the ChuA-αβHb complex by crystallography. Crystallisation screening was performed with 2:1 molar ratio of αβHb to ChuA, with pink/brown crystals forming in several conditions. X-ray diffraction data was collected

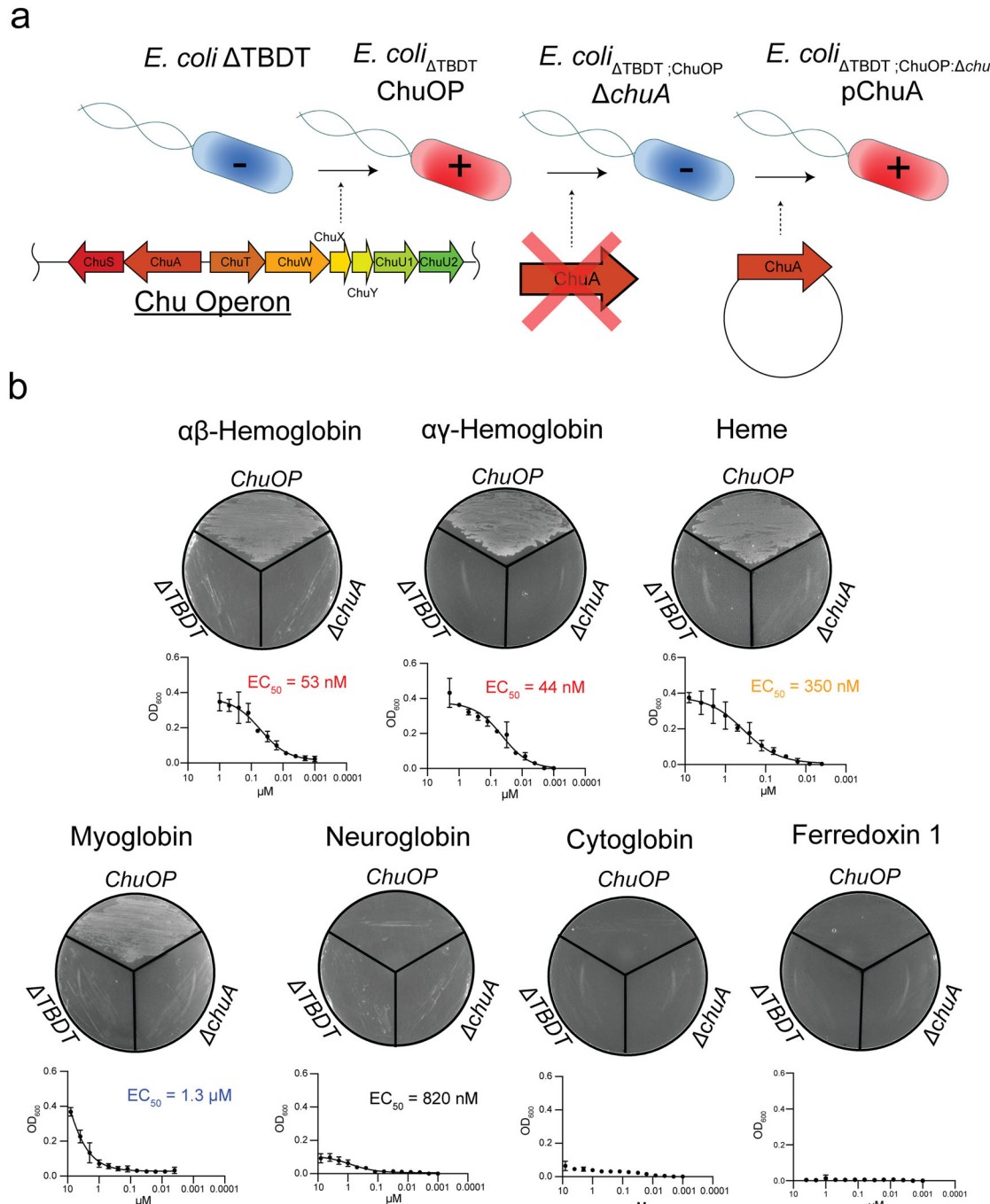

**Fig. 1 | ChuA targets hemoglobin, myoglobin and free hemin. a** Schematic of the genetic engineering strategy for the generation of the ChuA reporter strain used in this study. **b** The growth phenotype of the strain described in (**a**), grown on LB agar (top; representative images shown from $n = 3$, biological replicates) in the presence of iron-limited LB agar supplemented with either 5 μM αβHb, αγHb, hemin, myoglobin, neuroglobin, cytoglobin or human ferredoxin 1. (below) EC50 values of

*E. coli_{ΔTBDT;ChuOP}* cultured in LB liquid medium in the presence of 200 μM 2,2′-bipyridine, supplemented with serially diluted αβHb, αγHb, hemin, myoglobin, neuroglobin, cytoglobin or ferredoxin 1. Text colour for the EC50 values denotes the potency of the growth stimulatory effect (red = < 100 nM, orange = < 500 nM, black = < 1 μM, blue = > 1 μM). Data ($n = 3$, biological replicates) displayed as mean ± standard deviation.

from these crystals and solved by molecular replacement (Supplementary Table 2, and Supplementary Fig. 2a–c). However, only ChuA was observed in the crystal, with heme bound to His-420, located in extracellular loop 7 of the transporter (Fig. 2a). Aside from bound heme, the overall structure was highly similar to the previously solved apo-structure of the nearly identical transporter ShuA[17] (>99% AA identity, ChuA to ShuA substitutions V61I, and D234E; RMSD = 0.430 Å out of 3626/4638 atoms). The presence of heme bound to ChuA is consistent with previous work showing that the transporter forms a

transient interaction with αβHb, extracting heme, which binds at His-420[16]. We also attempted to determine the ChuA-αβHb complex by CryoEM, but only recovered class averages containing free αβHb and ChuA, further indicating that this interaction is transient (Supplementary Fig. 3).

To gain insight into the structure of the ChuA-hemoglobin complex we performed AlphaFold2 multimer modelling[34], providing the sequence of ChuA and one or two copies of the α and β Hb subunits, corresponding to a hemoglobin dimer or tetramer. The oligomeric

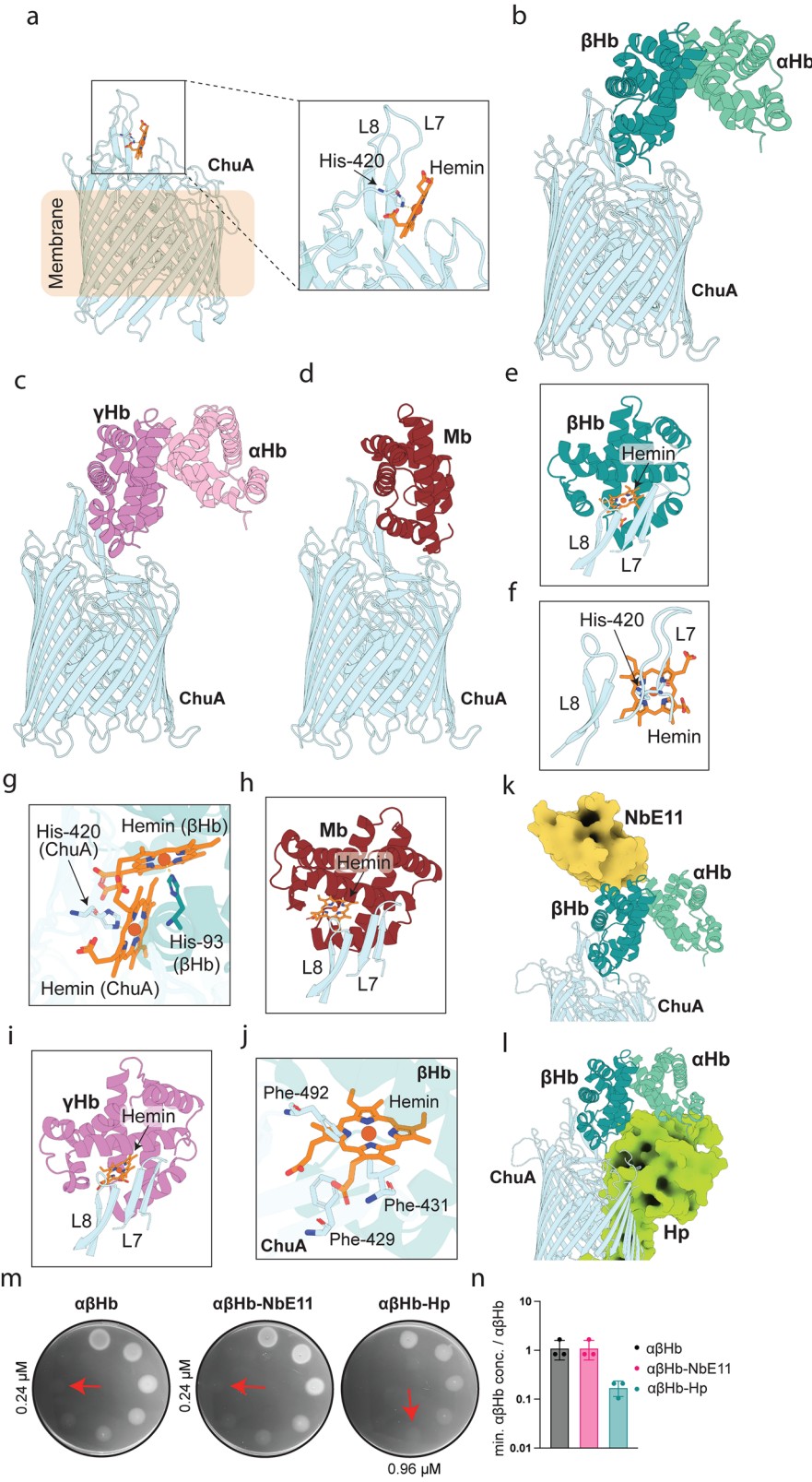

state of hemoglobin is pH and redox-state dependent (Supplementary Fig. 4). The dimeric form predominates in serum and thus is more likely to be relevant as an iron source for invading pathogens[16,22,35,36]. Consistent with this, while ChuA-αβHb$_{tetramer}$ modelling did not produce any realistic solutions, the top-ranked solution for the ChuA-αβHb$_{dimer}$ modelling placed the β-subunit of the Hb dimer in contact with extracellular loops 7 and 8 of ChuA, with reasonable confidence

scores (Fig. 2b, Supplementary Fig. 5a). We also performed AlphaFold2 modelling with dimeric αγHb and monomeric Mb, with these substrates predicted to form a complex with ChuA in the same orientation to the βHb subunit, adding to our confidence in this prediction (Fig. 2c, d, and Supplementary Fig. 5a). When hemin was appended to the βHb subunit in complex with ChuA, based on the crystal structure of hemoglobin (PDB ID: 2HHB, Fig. 2e)[37], its iron centre is only 8.5 Å

**Fig. 2 | ChuA coordinates heme at His-420 and binds hemoglobin and myoglobin via extracellular loops 7 and 8. a** Crystal structure of ChuA (light cyan) bound to hemin (orange) extracted from αβHb, showing the coordination of hemin by His-420 of loop 7. **b** The top-ranked AlphaFold2 multimer model of ChuA-αβHb$_{dimer}$ (αHb shaded light green, βHb shaded dark green). **c** The top-ranked AlphaFold2 multimer model of ChuA-αγHb$_{dimer}$ (αHb shaded pink, γHb shaded purple). **d** The top-ranked AlphaFold2 multimer model of ChuA-Mb (Mb shaded dark red). **e** Loop 7 and 8 of ChuA binding to βHb from the ChuA-αβHb$_{dimer}$ model, with hemin coordinated in βHb appended the crystal structure of Hb (PDB ID: 2HHB). **f** Loop 7 and 8 of ChuA bound to hemin, via interactions with His-420, from the ChuA-hemin crystal structure (**g**) Hemin coordinated in βHb by His-93 as in the ChuA-αβHb$_{dimer}$ model, overlayed with the coordination of hemin bound to ChuA by His-420 in the crystal structure. **h** Loop 7 and 8 of the ChuA-Mb model, showing hemin coordinated by Mb appended from the crystal structure of Mb (PDB ID: 3RGK). **i** Loop 7 and 8 of ChuA binding to γHb from the ChuA-αγHb$_{dimer}$ model, showing hemin coordinated in γHb appended from the crystal structure of αγHb (PDB ID: 4MQJ). **j** ChuA sidechain clashes with the hemin in βHb as shown in panel b. **k** Superimposition of αβHb-NbE11 (PDB ID: 8VYL) with the ChuA-αβHb$_{dimer}$ model, with NbE11 shown in surface view. **l** Superimposition of αβHb-Hp (PDB ID: 4WJG) with the ChuA-αβHb$_{dimer}$ model, with Hp shown in surface view. **m** Representative images of soft agar overlay assays (n = 3) of E. coli$_{\Delta TBDT:ChuOP:\Delta chuA}$ complemented with WT ChuA on iron-limited agar spotted with 2-fold serially-diluted αβHb, αβHb-NbE11 or αβHb-Hp. Minimal concentration supporting growth as indicated by red arrows. **n** Quantification of the minimal growth concentrations of αβHb, αβHb-NbE11 or αβHb-Hp, as a ratio of αβHb minimal growth concentration across three biological replicates (n = 3) displayed as mean ± s.e.m.

from that of bound heme in the ChuA crystal structure (Fig. 2f, g). Given their similar binding modes, heme present in both the γHb and Mb subunits were in a similar position (Fig. 2h, i). βHb in complex with heme at this position has several clashes with sidechains in loops 7 and 8 of ChuA (Fig. 2j), which may be involved in destabilising bound heme, facilitating its transfer to the proximal heme binding residue His-420, providing a plausible mechanism for heme extraction by ChuA.

To validate the ChuA-αβHb$_{dimer}$ AlphaFold model, we assessed the ability of ChuA to extract and import heme from αβHb in complex with dimeric human haptoglobin (αβHb-Hp)[38] and the hemoglobin binding nanobody NbE11 (αβHb-NbE11)[39] using an iron-limited agar plate growth assay. Based on the ChuA-αβHb$_{dimer}$ model and the crystal structures of αβHb-Hp and αβHb-NbE11 (PDB IDs = 4WJG, 8VYL)[37,39], the binding of NbE11 should not affect heme extraction by ChuA, while clashes between haptoglobin and ChuA should have a negative effect on heme extraction from αβHb-Hp (Fig. 2k, l). Consistent with this structural data, ChuA was equally effective at extracting heme from αβHb-NbE11 as αβHb, with a minimal growth concentration of 0.24 μM. Conversely, ChuA was significantly less effective at extracting heme from αβHb-Hp with a minimal growth αβHb concentration of 0.96 μM (Fig. 2m, n). The ability of ChuA to extract heme from αβHb-Hp, despite the significant clashes observed in the static modelling of this complex, is likely due to the flexibility of binding loops 7 and 8, combined with the dynamic nature of the interaction.

## His-420 and His-86 are required for growth on hemoglobin and hemin

To further validate the AlphaFold2 model of the ChuA-αβHb$_{dimer}$ and to gain insight into heme transfer, we complemented E. coli$_{\Delta TBDT:ChuOP:\Delta chuA}$ with either WT ChuA, a panel of ChuA mutants, or an empty vector control, and tested their ability to grow on αβHb. We utilised ChuA$_{H420A}$, ChuA$_{H86A}$, ChuA$_{H68A}$, single mutants to probe the role of possible heme coordinating histidine residues, as well as ChuA$_{I423A;N429A;F484A}$, ChuA$_{H420A;I423A;N429A;F484A}$ and ChuA$_{V482F;D483E;A485E}$ multiple mutants, based on the interaction interface of the αβHb binding site in our AlphaFold2 model (Fig. 3). We used iron-limited agar growth assays, spotted with serially diluted αβHb to determine the minimum concentration of αβHb that could support the growth of the complemented strains compared to WT ChuA (Figs. 3 and 4). E. coli$_{\Delta TBDT:ChuOP:\Delta chuA}$ complemented with wildtype ChuA has a minimal αβHb concentration of 0.24 μM, with no growth observed for the empty vector control (Fig. 3a, b). The ChuA$_{I423A;N429A;F484A}$ mutant, designed to stabilise interactions with αβHb (Fig. 3c), grew at a minimal Hb concentration of 0.48 μM αβHb, whereas the ChuA$_{V482F;D483E;A485E}$ mutant, designed to disrupt αβHb interactions by steric or electrostatic hindrance, grew at a minimal concentration of 0.24 μM, the same as wildtype (Fig. 3d). It is unclear why these mutants had a relatively minor or no effect on αβHb utilisation, given they cause changes at the predicted ChuA-αβHb interface. However, this may result from the dynamic nature of this interaction.

Both the single and multiple mutants containing the H420A substitution (ChuA$_{H420A}$, ChuA$_{H420A;I423A;N429A;F484A}$), did not grow at any αβHb concentration, consistent with previous reports that this residue is critical for heme extraction (Fig. 3e, f)[16]. The ChuA$_{H86A}$ mutant also had a significantly abrogated growth phenotype, with growth only observed at a αβHb concentration of 7.69 μM, confirming previous reports that this residue is also important for heme extraction (Fig. 3g)[16]. ChuA$_{H68A}$ exhibited similar growth to wildtype, with a 0.24 μM minimal αβHb concentration (Fig. 3h), indicating it is not likely to be involved in heme extraction or coordination.

To assess differences in the importance of ChuA binding site residues for heme binding and import, rather than heme extraction from hemoglobin (Fig. 4a), we repeated the growth experiments with free hemin (Fig. 4b). ChuA$_{I423A;N429A;F484A}$, and ChuA$_{H68A}$ mutants grew at minimal hemin concentrations equivalent to wildtype (6.1 μM), indicating these substitutions do not affect hemin binding or import (Fig. 3a, c, h). ChuA$_{V482F;D483E;A485E}$ exhibited a two-fold increase in the minimum hemin concentration for growth (12.2 μM) (Fig. 3d). Interestingly, at higher hemin concentrations (>191.7 μM) we observed growth inhibition due to excessive hemin import, likely due to the production of toxic hydroxyl radicals via Fenton chemistry or metalloporphyrin toxicity (Fig. 3a, c, d, h)[40] Analogous to growth on αβHb, the ChuA$_{H86A}$ mutant was functional, but impaired in its ability to utilise hemin, with growth observed to 96.6 μM (Fig. 3g). The ChuA$_{H420A}$ mutant was barely functional for hemin import with weak growth observed at the highest hemin concentration (773 μM) (Fig. 3f). Intriguingly, the additional substitutions in the ChuA$_{H420A;I423A;N429A;F484A}$ mutant partially restored ChuA function with a minimal hemin growth concentration of 48.3 μM (Fig. 3e). These additional mutations replace bulky residues surrounding His-420 with alanine, possibly relieving steric hindrance and allowing hemin binding to His-86, facilitating import.

To gain further insight into heme coordination by ChuA, we performed AlphaFold3 modelling of the ChuA-heme complex (Supplementary Data 1)[41]. Heme modelling varied across the five ranked models produced by AlphaFold3, ranging from coordination analogous to the ChuA-heme crystal structure with Fe coordination only by His-420, to bidentate coordination by both His-420 and His-86. This His-420 and His-86 coordination was facilitated by significant movement of ChuA loops 7 and 8 towards the plug domain (Fig. 4c). When the modelling was performed with either the H420A or H86A variant ChuA, heme was coordinated by the remaining histidine in a single binding mode (Fig. 4d). While the confidence scores for heme in these models were modest (Supplementary Fig. 5b, c), given the experimental evidence supporting the role of both His-420 and His-86 hemin coordination, it is likely that this modelling provides a realistic picture of heme coordination and dynamics by ChuA.

## De novo-designed ChuA binders block heme uptake from hemoglobin

Considering the importance of heme acquisition by pathogenic E. coli during infection[27,28,42], we sought to leverage our understanding of

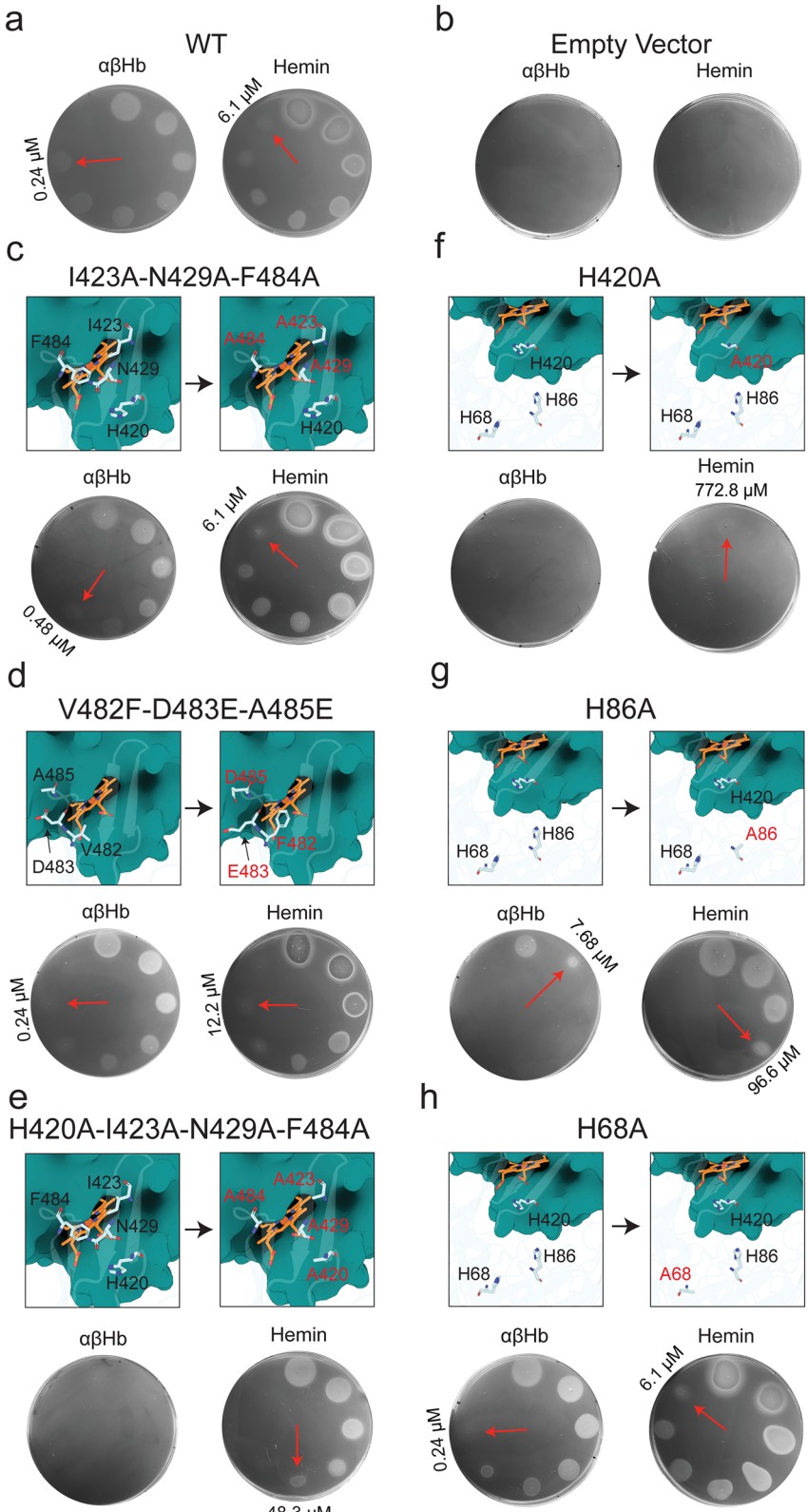

**Fig. 3 | ChuA His-420 and His-86 are required for growth on Hb.** Representative images of soft agar overlay assays ($n = 3$ biological replicates) of *E. coli*$_{\Delta TBDT:ChuOP:\Delta chuA}$ complemented with either WT ChuA (**a**), empty vector (**b**), ChuA$_{I423A,N429A,F484A}$ (**c**), ChuA$_{V482F,D483E,A485E}$ (**d**), ChuA $_{H420A,I423A,N429A,F484A}$ (**e**), ChuA $_{H420A}$ (**f**), ChuA $_{H86A}$ (**g**), or ChuA $_{H68A}$ (**h**), grown on iron-limited LB agar spotted with 2-fold serially-diluted αβHb or hemin. Minimal concentration supporting growth as indicated by red arrows. Inset displays WT and mutated residues as shown as sticks in the ChuA-αβHb$_{dimer}$ AlphaFold2 multimer model, with βHb shown in an overlaid cartoon-surface view.

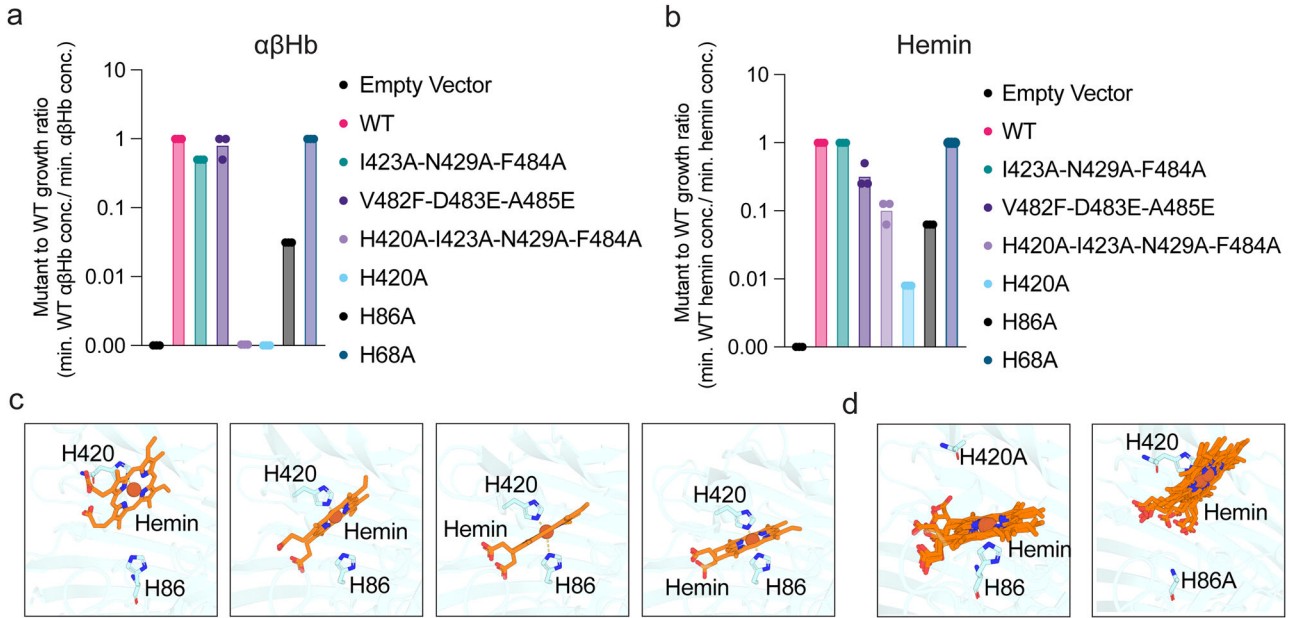

**Fig. 4 | ChuA His-420 and His-86 coordinate heme.** Quantification of the ratio of the minimal αβHb (**a**) and hemin (**b**) concentration supporting growth of *E. coli* ChuA mutant vs WT strains. Data represents three biological replicates (*n* = 3) displayed as mean ± s.e.m. **c** AlphaFold3 models of WT ChuA in complex with heme, suggesting that heme is coordinated by both His-420 and His-86, facilitated by the flexibility of ChuA loops 7 and 8. **d** AlphaFold3 models of ChuA H420A and H86A mutants in complex with heme, with all models predicting heme coordination by the remaining histidine residue.

ChuA function to design de novo protein binders to block heme acquisition from hemoglobin. Successful binder design validates both our model for hemoglobin binding by ChuA and provides a proof of concept for inhibiting *E. coli* infection by denying access to heme from host hemoglobin. Using an AlphaFold2 model of ChuA as a target, we utilised RFdiffusion and ProteinMPNN to design binders targeting extracellular loops 7 and 8 of ChuA, which our model indicated were responsible for hemoglobin binding (Fig. 2)[29,43,44]. We generated ~20,000 ChuA binders in silico and selected 96 designs for wet lab screening using AlphaFold2 filtering and manual curation[34,45]. We expressed and partially purified these ChuA binders in parallel on a small scale, with most binders expressing at similar levels (Supplementary Fig. 6a). We screened these binders for their ability to inhibit the growth of *E. coli*$_{ΔTBDT:ChuOP}$, by spotting them on hemoglobin agar overlaid with the bacteria. We observed weak zones of growth inhibition for many of the binders, suggesting a relatively high level of success at generating lower-affinity binders, with eight binders displaying more prominent zones of inhibition (Fig. 5a). Zones of inhibition were more prominent at a lower hemoglobin concentration, suggesting the binders act competitively (Supplementary Fig. 6b). Interestingly, no inhibition was observed on agar containing free hemin suggesting binders block heme-extraction from hemoglobin but not heme transport by ChuA (Supplementary Fig. 6c).

We expressed and purified four binders with the most prominent inhibition zones (A10, C8, G7 and H3) (Fig. 5d, Supplementary Fig. 6d, Supplementary Data 2 and 3). All binders inhibited αβHb and Mb-dependent growth of *E. coli*$_{ΔTBDT:ChuOP}$ when spotted on agar, to a concentration of 250-500 nM for αβHb and 125-250 nM for Mb (Fig. 5b,c). There was no inhibition when hemin or Fe(II)SO$_4$ was provided as the iron source (Supplementary Fig. 6e, f). The higher potency of binders with Mb as the heme source is likely due to the lower affinity of ChuA for this protein. To more robustly quantify binder potency, we determined binder IC$_{50}$ for *E. coli*$_{ΔTBDT:ChuOP}$ in liquid culture supplemented with either 50 nM or 100 nM αβHb. At 100 nM αβHb, IC$_{50}$ values ranged from 3.3 μM for A10 to 42.5 nM for G7 (Fig. 5e). Lower IC$_{50}$ values were obtained at 50 nM αβHb consistent with competitive inhibition for the ChuA binding site (Supplementary Fig. 6g). Next, we

determined the affinity of binders A10, G7, and H3 for purified ChuA using BLI, with average disassociation constants of 127, 84.9 and 71.4 nM recorded for A10, G7, and H3 respectively (Fig. 5f, Supplementary Table 1), which are broadly consistent with growth inhibition values. The association ($k_{on}$) and dissociation rate ($k_{off}$) constants varied considerably between the binders (Supplementary Table 1, Supplementary Fig. 6h), with A10 forming a more transient interaction with ChuA, which may explain its higher IC$_{50}$ value, despite its roughly comparable K$_D$ to G7 and H3. To test binder specificity for ChuA, we repeated the BLI experiment with the purified *E. coli* proteins FhuE and YddB (unrelated TBDTs), and PqqL (periplasmic protease)[30,32,46]. While characteristic binding curves were observed with ChuA, no binding signal was observed for any other protein, indicating that ChuA binders are specific for their design target (Supplementary Fig. 6i).

## ChuA-binder complex structures closely match the computational design

Our functional and biochemical analysis of the de novo-designed ChuA binders definitively demonstrates they are high-affinity inhibitors of heme extraction from hemoglobin. To resolve how closely their computational design matches reality, we attempted to determine the crystal structures of A10, C8, G7 and H3 in isolation. Diffracting crystals were only obtained for binder C8, and the structure was solved at 2.46 Å (Supplementary Table 2). The crystal contained eight molecules per asymmetric unit, which have a pairwise similar RMSD of ~0.6 Å (Supplementary Fig. 7a, b). This was comparable to the computational design of C8, which also had an RMSD of ~0.6 Å to the C8 molecules in the crystal structure (Supplementary Fig. 7c, Supplementary Table 3), indicating that the predicted model closely matches the X-ray structure, once flexibility of the protein and experimental error are accounted for.

To determine if the similarity between the designed and experimental structure extends to the binders in complex with ChuA, we determined the CryoEM structures of G7 and H3 in complex with the transporter (Supplementary Fig. 2, Supplementary Table S4). The structures of both binders in complex with ChuA were resolved to < 3 Å, allowing us to model these complexes with high confidence

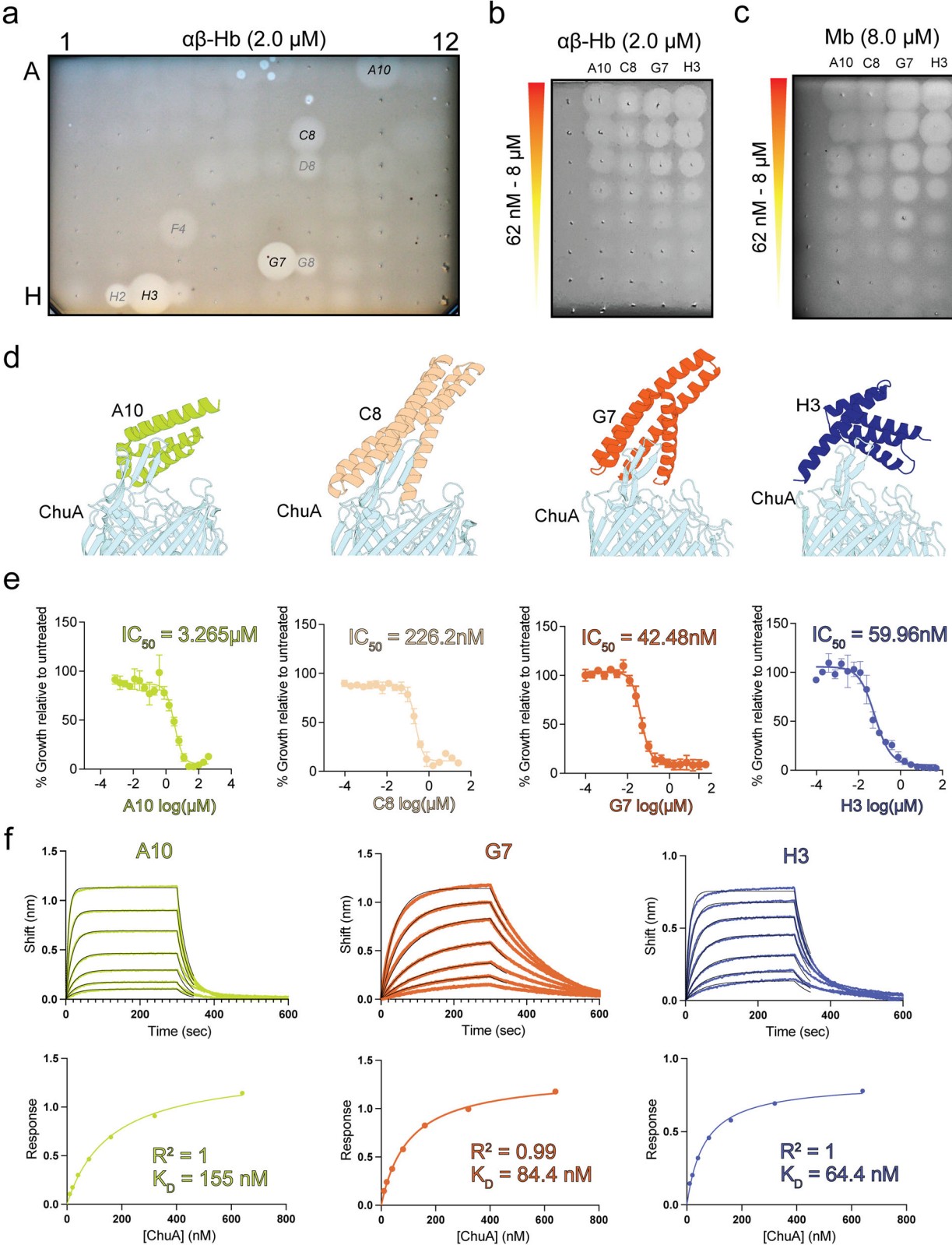

(Fig. 6a, b). These experimental structures were very similar to the predicted models, with the RMSD between the binders of ~0.6 Å (Fig. 6c, d). The position of H3 in the full complex differed slightly between the designed and experimental structures, due to a change in the conformation of ChuA loops 7 and 8 (Fig. 6d). However, as these loops are flexible both the design and experimental structure likely represent possible conformations in solution. These structures

provide compelling evidence that the de novo design of ChuA binding proteins closely matches experimental reality.

An overlay of the ChuA-binder structures, with the predicted ChuA-αβHb complex, shows that both G7 and H3 occupy the majority of the predicted αβHb binding surface of ChuA loops 7 and 8 (Fig. 6e, f), providing a clear mechanism for the observed inhibition of heme extraction by ChuA. Conversely, the same binder overlay with

**Fig. 5 | *De novo*-designed ChuA binders bind with high affinity, inhibiting heme extraction from hemoglobin. a** A representative image of a soft agar overlay assay (*n* = 3) of *E. coli*$_{\Delta TBDT:ChuOP}$, grown on iron-limited LB agar containing 2 μM αβHb, spotted with 2 μl of 96 purified the de novo ChuA binders. **b** A soft agar overlay assay of *E. coli*$_{\Delta TBDT:ChuOP}$, grown on iron-limited LB agar containing 2 μM αβHb, spotted with 2-fold serial dilution of de novo binders A10, C8, G7 and H3 (8 μM – 62 nM). **c** As in panel b but supplemented with 8 μM Mb instead of αβHb. **d** The top-ranked AlphaFold2 multimer model of ChuA bound to binders A10, C8, G7 or H3.

**e** Binder IC$_{50}$ of *E. coli*$_{\Delta TBDT:ChuOP}$ grown iron-limited LB broth containing 0.1 μM αβHb, with 2-fold serially diluted de novo binders A10, C8, G7 or H3. IC$_{50}$ values were calculated as a % relative to the growth of the untreated control. Data (*n* = 3) displayed as mean ± s.e.m. **f** Representative BLI sensorgram traces (top) and associated steady-state binding kinetics (bottom) of A10, G7 and H3 binding to ChuA. All BLI experiments were performed three times (*n* = 3), with comparable results. K$_D$ values represent the average of these three experiments.

the ChuA-heme experimental structure, shows that only minor clashes occur between the heme bound at His-420 and the binding proteins (Fig. 6g, h). Given that ChuA coordinates heme via both His-420 and His-86, in a process mediated by the flexibility of loops 7 and 8, and the fact that His-420 is conditionally dispensable for heme uptake, these structures are consistent with ChuA remaining capable of heme import even in complex with these binding proteins.

## Discussion

In this work, we show that ChuA is a heme transporter that targets dimeric hemoglobin (αβHb or αγHb) as its high-affinity substrate and is also capable of importing free hemin or heme from myoglobin. Considering most heme in the human body is contained within αβHb, and free heme is rapidly sequestered by hemopexin or serum albumin[22,47,48], these findings indicate that αβHb is likely the physiological substrate for ChuA. Further, considering αβHb in serum is rapidly and tightly bound by haptoglobin, and ChuA can to some extent utilise αβHb-Hp, it is likely this complex is also a physiological substrate for ChuA[49]. We also clarify the role of His-86 and His-420 in ChuA function, showing that while these histidines are important for heme extraction from αβHb, both are conditionally dispensable for the import of free hemin. This observation combined with our structural data indicating both His-86 and His-420 directly coordinate the hemin Fe centre, hints that heme extraction and import by ChuA is a dynamic multistage process. We leverage this understanding of ChuA function to generate de novo designed binding proteins, which selectively block heme extraction from αβHb by ChuA. Two of these binders achieved sub-100 nM affinity, without experimental optimisation, validating AI-guided protein design for binder generation against flexible extracellular regions of integral membrane protein targets. The success of our binder design supports our model for heme extraction and binding by ChuA, providing insight into this dynamic and difficult-to-study process, and demonstrating the utility of de novo-designed binders as research tools. Our structural analysis of the ChuA-G7 and Chu-H3 complexes by CryoEM shows that RFdiffusion-based computational design closely matches experimental reality and allows for a definitive explanation of the effect of these binders on ChuA function. Finally, the ability of these binders to inhibit *E. coli* growth provides a strong proof of concept of the use of de novo-designed binding proteins as antimicrobials, which block the uptake of essential nutrients at the cell surface.

## Methods
### Ethics Statement
This research complies with all relevant ethical regulations approved by the Monash University Human Research Ethics Committee

### Protein expression and purification
**ChuA.** WT ChuA lacking the signal peptide sequence was amplified from *E. coli* CFT073 by PCR, with primers containing NcoI and XhoI restriction sites to clone ChuA into a modified pET20b vector, in frame with an N-terminal PelB signal sequence, followed by a 10x His tag and a TEV protease cleavage site (Supplementary Data 4). pET20bChuA was then transformed into *E. coli* BL21 (DE3) C41 cells and plated onto LB agar supplemented with 100 μg/ml ampicillin and incubated at 37 °C overnight. An overnight starter culture was prepared in the same

medium, before being sub-cultured the next day in terrific broth supplemented with 100 μg/ml ampicillin at 37 °C. 8 L cultures were grown until an OD$_{600}$ of 0.8-1.0 was reached, and protein expression was then induced by the addition of 0.3 mM isopropyl 1-thio-b-D-galactopyranoside (IPTG), and were then grown for a further 16 h at 24 °C. Cells were harvested via centrifugation and pellets were homogenised in lysis buffer (50 mM Tris, 200 mM NaCl, pH 7.8, supplemented with 1 mM MgCl2, 0.1 mg/ml lysozyme, 0.05 mg/ml DNase 1 and x1 Complete protease inhibitor tablet (Roche)/8 L culture) and placed in an ice bath slurry for 30 minutes before being lysed with a cell disruptor (Emulseflex, one pass-through). Lysate was clarified via centrifugation at 20,000 g for 10 minutes at 4 °C and the supernatant was ultracentrifuged at 100,000 g for 1 h at 4 °C to isolate the membrane fraction. The membrane pellet was then solubilised in lysis buffer using a tight-fit dounce homogeniser before the addition of 5% Elugent (Santa Cruz Biotechnology) and incubated at room temperature with gentle rocking for 20 minutes. A 5 ml HisTrap nickel column was prepared via washing with 10-15 column volumes of MilliQ H2O followed by 10-15 column volumes of binding buffer (50 mM Tris, 500 mM NaCl, 20 mM imidazole, 0.03% dodecyl maltoside (DDM), pH 7.8) before binding loaded with solubilised membranes. The resin was washed with a further 10-15 column volumes of binding buffer before elution of protein fractions via a stepwise gradient (10%, 25%, 50%, 75% and 100%) in elution buffer (50 mM Tris, 500 mM NaCl, 1 M imidazole, 0.03% DDM, pH 7.8). Fractions were run via SDS-PAGE and analysed via Coomassie gel staining. Fractions containing ChuA were pooled and concentrated using a 100 kDa spin filter column (Millipore) before loading onto a S200 26/600 Superdex size exclusion chromatography (SEC) column (Cytiva) preequilibrated in SEC buffer (50 mM Tris, 200 mM NaCl, 0.03% DDM, pH 7.8) and were run using the AKTA Pure system (Cytiva). Fractions were then run via SDS-PAGE and analysed via Coomassie staining. Fractions containing ChuA were pooled and concentrated with a separate 100 kDa spin filter column before being stored at -80 °C. The average yield for ChuA purification was ~0.35 mg/L culture.

**Recombinant globins.** Sequences for the expression of recombinant human αβ and αγ hemoglobin, myoglobin, neuroglobin, cytoglobin and ferredoxin 1 were ordered as gene fragments from Twist Biosciences and cloned into either pETDuet-1 (for hemoglobin), pET29a or pET22b, carrying an N-terminal 6x His tag. For αβ and αγ hemoglobin, a 6x His tag was added to either the N-terminal of the alpha subunit or the C-terminal of the β or γ subunit in the event presence of the tag interfered with tertiary structure formation. Expression vectors were transformed into *E. coli* BL21 (DE3) C41 cells and plated onto LB agar supplemented with 100 μg/ml ampicillin and incubated at 37 °C overnight. An overnight starter culture was prepared in the same medium, before being sub-cultured the next day in terrific broth supplemented with 100 μg/ml ampicillin at 37 °C. 3-4 L cultures were grown until an OD$_{600}$ of 0.8–1.0 was reached, and protein expression was then induced by the addition of 0.3 mM IPTG and were then grown for a further 16 h at 22 °C. Cells were harvested and lysed in lysis buffer (50 mM Tris, 200 mM NaCl, pH 7.4, supplemented with 1 mM MgCl$_2$, 0.1 mg/ml lysozyme, 0.05 mg/ml DNase 1 and x1 Complete protease inhibitor tablet (Roche)/8 L culture) and placed in an ice bath slurry for 30 minutes before being lysed with a

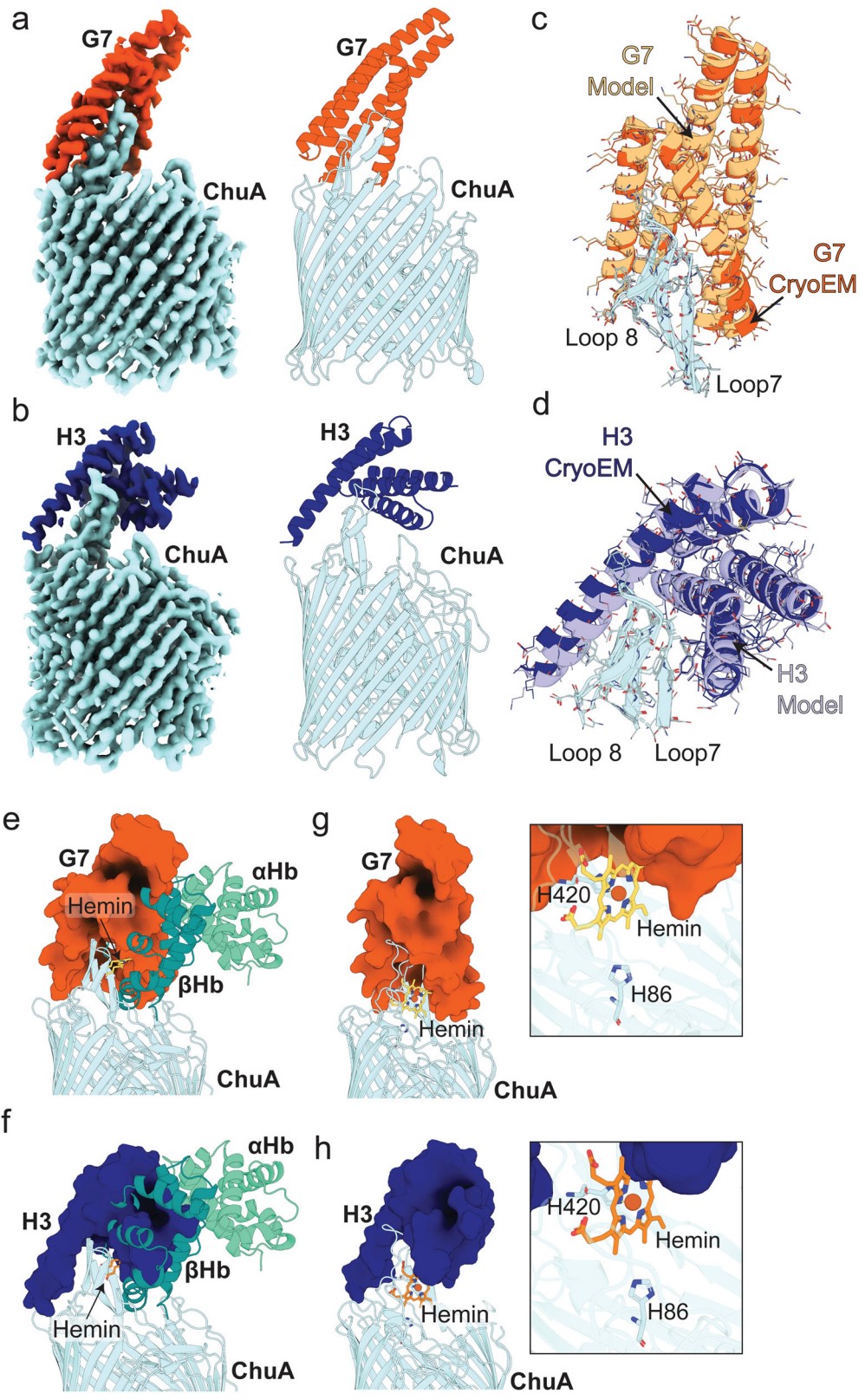

cell disruptor (Emulseflex, one pass-through). Lysate was clarified via centrifugation at 20,000 g for 10 minutes at 4 °C before loading into a HisTrap nickel column, pre-equilibrated in binding buffer (50 mM Tris, 500 mM NaCl, pH 7.4). Fractions were run via SDS-PAGE and analysed via Coomassie gel staining. Fractions containing purified globins were pooled and loaded onto a S75 26/600 Superdex SEC column (Cytiva) using a 50 ml Superloop (Cytiva) preequilibrated in

SEC buffer (50 mM Tris, 200 mM NaCl, pH 7.4) and were run using the AKTA Pure system (Cytiva). Fractions were then run via SDS-PAGE and analysed via Coomassie staining. Fractions containing purified recombinant globins were pooled and concentrated with a separate 3 or 10 kDa spin filter column before being stored at −80 °C. The average yield for these purifications was: αβHb α n-term 6xhistag: ~6 mg/L; αβHb β c-term 6xhistag: ~25 mg/L; αγHb α n-term 6xhistag:

**Fig. 6 | The cryoEM structures of ChuA-G7 and ChuA-H3 closely match the computational design and block hemoglobin binding via steric hindrance.** The final cryoEM density maps of the ChuA-G7 complex (**a**) or ChuA-H3 complex (**b**) (ChuA shaded cyan, G7 shaded dark orange, H3 shaded dark blue) (left) and the refined structures of the ChuA-G7 complex or the ChuA-H3 shown as a cartoon representation (right). Zoomed view of G7 (**c**) or H3 (**d**) bound to loops 7 and 8 of ChuA, with the cryoEM structure overlaid (G7 shaded dark orange, H3 shaded dark blue) with the AlphaFold2 model (G7 shaded light orange, H3 shaded light blue), showing the structures of ChuA-G7 and ChuA-H3 closely match the computation designs. The structure and model are depicted as cartoon representation, with sidechains shown as thin sticks. Overlay of the ChuA-G7 (**e**) or ChuA-H3 (**f**) complex structures with the ChuA-$\alpha\beta$Hb$_{dimer}$ AlphaFold2 multimer model, showing binders block the predicted ChuA $\alpha\beta$Hb binding site, preventing heme extraction. Overlay of the ChuA-G7 (**g**) or the ChuA-H3 complex (**h**) structures with the ChuA-heme crystal structure, showing only minor clashes with heme bound H420, and G7 or H3, and the area surrounding is H86 unaffected.

---

~5 mg/L; $\alpha\gamma$Hb γ n-term 6xhistag: ~14 mg/L; Mb: ~80 mg/L; Nb: 60 mg/L; Cb: ~30 mg/L; Fd1: 12 mg/L of culture.

**Hemoglobin nanobody NbE11.** The human hemoglobin nanobody NbE11 was expressed and purified as described previously[39]. The DNA encoding the NbE11 gene was synthesised by (Twist Biosciences) and cloned into pET29a at the corresponding NcoI and XhoI restriction sites, such that the protein was encoded with an N-terminal His$_6$-tag. This construct was then used to express protein for structural and biochemical studies. pET29a-NbE11 was transformed into *E. coli* C41 (DE3) and cells were grown to OD$_{600}$ ~1 in terrific broth (12 g of tryptone, 24 g of yeast extract, 12.3 g of K$_2$HPO4, 2.31 g of KH$_2$PO4, 12.7 ml of glycerol) before protein expression was induced with 0.3 mM isopropyl 1-thio-β-D-galactopyranoside. Cells were then further cultured for 16 h at 24 degrees. The average yield from this purification was ~60 mg/L culture.

**Purification of hemoglobin from erythrocytes.** Hemoglobin was purified from human blood collected from a consenting, healthy adult volunteer in accordance with 2022-30658-70864 approved by the Monash University Human Research Ethics Committee. Erythrocytes were isolated via centrifugation and cell pellets were diluted in 50 mM Tris, 200 mM NaCl, pH 7.4. To isolate hemoglobin, Erythrocytes were lysed with a tight-fit dounce homogeniser and clarified by centrifugation at 30,000 g. Cell lysate was further purified using a Superdex 200 10/300 column (Cytiva) preequilibrated in 50 mM Tris, 200 mM NaCl, pH 7.4. Fractions were collected and pooled and concentrated to ~10 mg/ml using a 30 kDa spin filter column (Millipore). The average yield for this purification was ~150 mg/mL of erythrocytes.

**Generation of different hemoglobin oligomeric and redox states** To generate different oligomeric states of hemoglobin, 500ul of hemoglobin purified from human erythrocytes was diluted to 1.5 ml in 50 mM BisTris, 100 mM NaCl, at either pH 6.0, 6.5, 7.0, or 8.0. The hemoglobin samples were run through pre-equilibrated a HiTrap desalting column (Cytiva) and the flow through containing hemoglobin was collected. The flow-through was applied to the HiTrap desalting column three times to ensure full buffer exchange. Hemoglobin samples were left to incubate at room temperature in buffer for 30 minutes before loading 5 µM onto a pre-equilibrated S200 10/300 column for analytical SEC analysis. The mass of hemoglobin in buffer at either pH 6.0 or 8.0 was further confirmed using a Refeyn 2MP Mass Photometer.

To generate methemoglobin (metHb) from oxyhemoglobin (oxyHb), oxyHb was incubated with NaNO$_2$ at a 1:5 MR at room temperature shaking for 2 h, as described previously[36]. The samples were then buffer exchanged into 50 mM Tris, 200 mM NaCl, pH 7.4 to remove excess NaNO$_2$. The absorbance spectra of NaNO$_2$-treated and untreated samples were measured using a Tecan Infinite 200 PRO microplate reader between 500-750 nm, and normalised at 500 nm, to identify the presence or absence of distinctive peaks corresponding to each of the Hb redox states[50].

**Genetic engineering of ChuA mutants** Mutant genotypes of *E. coli* BW25113 were generated using the λ-red system as described previously[51]. The *E. coli*$_{ΔTBDT}$ strain lacking all native TonB-dependent transporters; deleted in the following

sequence: *fhuA, fecA, cirA, fepA, fhuE, fiu, yncD,* and *yddB, was used as the base strain*[30–32,52]. The *E. coli*$_{ΔTBDT}$ strain was then recombineered with the entire Chu operon (Fig. 1a) to generate *E. coli*$_{ΔTBDT:ChuOP}$. DNA encoding the Chu operon, +500 bp up- and downstream, was amplified by PCR from the uropathogenic *E. coli* strain CFT073. 100 µg of this PCR product was electroporated into *E. coli*$_{ΔTBDT}$ containing the λ-recombinase containing plasmid pKD46 induced with arabinose. Cells were plated onto LB-agar containing 150 µM 2'2-bipyridine and 1 µM $\alpha\beta$Hb. Emergent colonies were screened by PCR to confirm integration of the Chu operon into the *E. coli*$_{ΔTBDT}$ genome. The *chuA* gene was then deleted from *E. coli*$_{ΔTBDT:ChuOP}$ strain using the λ-red system to generate *E. coli*$_{ΔTBDT:ChuOP:ΔchuA}$. *E. coli*$_{ΔTBDT}$ and all derived strains required additional iron for growth, which was provided by the supplementation of LB broth with 2.5 nM Fe(II)SO$_4$ at 37 °C.

**Bacterial agar growth assays**
To determine whether the ChuA strains could utilise different sources of heme or iron, each strain was streaked onto different LB agar plates containing 150 µM 2'2-bipyridine (BP; an iron chelator, to control for any iron present in the LB), supplemented with 5 µM sterile human $\alpha\beta$ or $\alpha\gamma$ hemoglobin, myoglobin, cytoglobin, neuroglobin or ferredoxin 1. Plates were then incubated at 37 °C overnight.

**Complementation of *E. coli*$_{ΔTBDT:ChuOP:ΔchuA}$ with WT or mutant ChuA**
Sequences for the 6 ChuA mutants, in addition to WT ChuA, were ordered from Twist Bioscience and were amplified and cloned into pTET (originally from Ben Adler at UC Berkeley) using GoldenGate cloning using the BbsI restriction sites. The resultant vectors (pTET WT or mutant ChuA) were sequenced at Primordium Inc. and were confirmed to carry the correct sequences. pTET WT or mutant ChuA were electroporated into *E. coli*$_{ΔTBDT:ChuOP:ΔchuA}$ and were maintained on LB agar supplemented with 2.5 nM Fe(II)SO$_4$ and 34 µg/ml chloramphenicol at 37 °C. To test for complementation *E. coli*$_{ΔTBDT:ChuOP:ΔchuA}$:pTET WT or mutant ChuA were inoculated in soft agar overlay assays (base: 1.5% agar, supplemented with 100 µM BP, 200 ng/ml anhydrous tetracyline and 34 µg/ml chloramphenicol; soft agar overlay: 0.6% agar, supplemented with 100 µM BP, 200 ng/ml anhydrous tetracyline and 34 µg/ml chloramphenicol) at an OD$_{600}$ of 0.1. 1:2 Serially-diluted sterile hemoglobin, hemin, NbE11-Hb (MR 2:1 NbE11:Hb$_{tetramer}$) or Hp-Hb (MR 1:2 Hp$_{dimer}$:Hb$_{dimer}$), was spotted over the dry plates (with a top concentration of 0.5 mg/ml) and were subsequently cultured at 30 °C for two days.

**Serial dilution *E. coli* growth experiments**
**Globins.** To determine if ChuA preferentially acquires heme from different heme sources, *E. coli*$_{ΔTBDT}$, *E. coli*$_{ΔTBDT:ChuOP}$ and *E. coli*$_{ΔTBDT:ChuOP:ΔChuA}$ were grown in 5 mL of 2.5 nM Fe(II)SO$_4$-supplemented LB broth and incubated overnight at 37 °C. The next day, 10 ml of LB supplemented with 150 µM BP was inoculated with each genotype at OD$_{600}$ of 0.01. 100 µL of inoculated medium was then added to each well in a 96-well plate. 12 µM of $\alpha\beta$ or $\alpha\gamma$ hemoglobin or 48 µM of either hemin, myoglobin, cytoglobin, neuroglobin or ferredoxin 1 was then added to 200 µL of the inoculated medium in an Eppendorf tube, and 100 µL of this medium was then added to the first well and then serially diluted by a factor of 2. Plates were then incubated at 37 °C

overnight and the $OD_{600}$ was measured using a Tecan Infinite 200 PRO microplate reader.

**De novo binders.** To generate IC50s, 10 ml of LB was supplemented with 200 μM BP and 0.2 μM hemoglobin, and was inoculated with the $E. coli_{\Delta TBDT:ChuOP}$ strain to an initial $OD_{600}$ of 0.01. In a 96-well plate, binders were serially diluted 1:2 for 24 dilutions, with a top concentration of 400 μM, and the plates were wrapped in parafilm before incubation at 37 °C at 100 rpm. The $OD_{600}$ was read 24 h later using a ClarioStar microplate reader.

ChuA binders were also serially diluted onto soft agar overlay assays (base: 1.5% agar, supplemented with 100 μM BP; soft agar overlay: 0.6% agar, supplemented with 100 μM BP, inoculated with $E. coli_{\Delta TBDT:ChuOP}$ at a starting $OD_{600}$ of 0.1, with either 2 μM hemoglobin, 8 μM myoglobin, 8 μM hemin or 2.5 nM Fe(II)SO₄, with the latter without any BP added to either the base or soft overlay). A 1:2-fold dilution series of ChuA binders A10, C8, G7 and H3 (with a top concentration of 8 μM) were spotted over the dry plates and were subsequently cultured at 37 °C overnight.

### AlphaFold2/3 modelling
AlphaFold2 multimer modelling predictions were run using AlphaFold version 2.1.1[34] through the Monash MASSIVE M3 computing cluster. Modelling of ChuA with heme using AlphaFold3 was conducted using the AlphaFold Server[41]. The top five ranked models were interrogated, and the top-ranked model was visualised using PyMOL (Schrödinger). The following structures were used as overlay in the models for the appropriate placement of heme in hemoglobin (PDB ID: 2HHB)[53] and myoglobin (PDB ID: 3RGK)[54], or for superimposition of the αβHb-Hp complex (PDB ID: 4WJG)[39].

### Design of de novo binders by RFdiffusion-ProteinMPNN-AF2 initial guess
A search model was prepared for binder backbone generation in RFdiffusion[29], using an AlphaFold2 model of ChuA, defining the outer loops of the barrel and plug domain and excluding the remainder of the model from the search (ChuA search amino acids 51–61, 66–70, 81–89, 175–193, 218–246, 272–291, 321–340, 366–381, 406–442, 474–499, 523–539, 564–579, 609–621). This model was provided to RFdiffusion as a target, with amino acids 572, 594, 563, and 568 defined as hotspots. Binder size was specified as 70-90 amino acids. The denoiser noise scale and scale frame were 0, and standard model weights were used. A total of 5,000 models generated in RFdiffusion, were provided to the DL binder design pipeline[44], for sequence assignment using ProteinMPNN[43] and quality screening using AlphaFold2 initial guess. Binders were screened for in silico success, using an AlphaFold2 initial guess pAE interact score cutoff of < 10. The RFdiffusion output for successful designs (~200 models) where then recycled through the DL binder design pipeline five times, to generate a final pool of models for selection (~550 models) with a pAE interact score cutoff of < 10. These models were further screened by complex prediction with full-length ChuA using AlphaFold2 multimer 2.3.2[34,55] with the 200 models with the highest pAE interact scores selected. These models were manually curated to maximise structural variability (length, helix topology, α and β secondary structure), to maximise binder-ChuA interaction interface area, and based on suitability for affinity tagging, to identify 96 designs for synthesis and testing.

### Expression and purification of de novo binders
Binder gene sequences were synthesized by Twist Bioscience and inserted in a pET29b expression vector between NdeI and XhoI binding sites. A N or C-terminal 6xHis tag was utilised depending on which of the binder termini was not involved in interactions with ChuA. Initial binder expression was performed in parallel. Binder expression plasmids were transformed into *E. coli* C41 DE3 cells by heat shock in a 2 ml

96 well plate (0.5 μl plasmid DNA at 10-50 μg/ml, 10 μl chemically competent *E. coli* cells). 1 ml of LB broth was added to cells before recovery at 37 °C for 1 h, with shaking at 400 rpm. Kanamycin selection was then added (50 μg/ml), and cells were incubated overnight at 37 °C. The binder construct transformed *E. coli* C41 DE3 cells were used to inoculate 4 ml of overnight express terrific broth (Merck) with 50 μg/ml Kanamycin in 24 well plate format. Cultures were grown for 18 h at 30 °C before the cells were harvested via centrifugation. The supernatant was eluted and cells were lysed using B-PER reagent (ThermoFisher Scientific) following the manufacturer's instructions. Lysed cells were clarified by centrifugation and the supernatant was transferred to a 2 ml 96 well plate, and 50 μl of Ni-agarose resin slurry was added, before incubation with shaking at 200 rpm for 1.5 h. The cell lysate was then added to a 96-well filter plate (30-40 μM cutoff), and lysate was removed with a vacuum manifold retaining the resin on the filter. Wells were washed with 3 × 1.5 ml of wash buffer (50 mM Tris, 500 mM NaCl, 20 mM imidazole, pH 7.8), before binders were eluted by adding 2 × 200 μl of elution buffer (50 mM Tris, 500 mM NaCl, 500 mM imidazole, pH 7.8), into a separate 500 μl 96 well plate. Binder expression and purity were assessed by SDS-PAGE. Larger scale production of binders A10, C8, G7, and H3 was performed as described above for the recombinant globins. The average yield for the binding proteins ranged from 50–200 mg/L of culture.

### Screening of de novo binders
For screening of the putative de novo binders, the $E. coli_{\Delta TBDT:ChuOP}$ strain in a soft agar overlay assay was used. The base agar (1.5%) was supplemented with 100 μM BP and was left to set. To this, a soft agar (0.6%) overlay was supplemented with 100 μM BP, 0.125 mg/ml (1.94 μM) hemoglobin and inoculated with the $E. coli_{\Delta TBDT:ChuOP}$ strain at an initial $OD_{600}$ of 0.1 and left to set. 2 μl of each of the 96 binders was spotted onto the soft agar overlay and plates were then incubated at 37 °C.

### X-Ray crystallography, data processing, refinement and analysis
**ChuA-heme.** Purified ChuA (10 mg/ml final concentration) in 0.8% β-octylglucoside was combined with purified human αβHb (~20 mg/ml final concentration) in a 1:2 ratio, and the complex was screened for crystallisation conditions (sitting drop, 100 nl protein + 100 nl crystallisation solution) using commercially available crystallisation screens (Index, JCSG +, MIDAS, PACT, ShotGunSG1 and PEG Ion) (Hampton Research, Molecular Dimensions). Pink-brown crystals formed in the PACT crystal screen condition containing: 0.1 M MIB buffer, 25% (w/v) PEG 1500, pH 6. Crystals for data collection were prepared from an optimisation grid screen ranging from 24–28 (w/v) PEG 1500, and pH 4-7, with the MIB buffer concentration constant. Excess mother liquor was removed by wicking, before being cryo-cooled in liquid N₂ at 100 K. Data was collected at the Australian Synchrotron, with crystals diffracting anisotropically to 2.8 Å (resolution ranged from 2.8 Å along the *k*-axis to 3.43 along the *l*-axis, judged by I/σ(I) = 1.5) in the $P2_12_12_1$ space group. The crystal structure of ShuA from *Shigella dysenteriae* (PDB ID = 3FHH)[38] was used for phasing via molecular replacement using Phaser[56]. Based on this solution we found that the crystals only contain ChuA in complex with heme that had been extracted from αβHb. The resulting structure was rebuilt manually and refined using PHENIX[57] and Coot[58].

**De novo ChuA binder C8.** Purified binder C8 (8 mg/ml) was screened for crystallisation conditions (sitting drop, 100 nl protein + 100 nl crystallisation solution) using commercially available crystallisation screens (Index, JCSG +, MIDAS, PACT, ShotGunSG1 and PEG Ion) (Hampton Research, Molecular Dimensions). Crystals with different morphologies formed in a range of conditions (Sheet/rectangular crystals formed in 0.2 M Na Acetate, 0.1 M Tris pH 8.5, 30% w/v PEG 4 K; Needle-like crystals formed in 0.2 M (NH₄)₂SO₄, 0.1 M Na Acetate pH

4.6, 30% w/v PEG MME 2 K; Hexagonal crystals formed in 0.2 M K Na Tartrate, 2 M $(NH_4)_2SO_4$, 0.1 M $Na_3$ Citrate pH 5.6). Crystals were looped directly from screening trays, and the excess mother liquor was removed before cryocooling in liquid $N_2$ at 100 K. Data was collected at the Australian Synchrotron. Most crystals diffracted poorly (highest resolution diffraction = 6-8 Å). However, a single crystal from the 0.2 M Na Acetate, 0.1 M Tris pH 8.5, 30% w/v PEG 4 K condition diffracted to ~2.5 Å. Data was collected on this crystal and the structure was solved by molecular replacement using an AlphaFold2 model of C8. The crystal contained 8 molecules per asymmetric unit, consisting of two groups of four molecules related by strong translational non-crystallographic symmetry (tNCS). The structure was rebuilt manually and refined using PHENIX[55] and Coot[58]. While the majority of all 8 molecules could be built into the available electron density, the refinement R-factors remained high due to the tNCS. However, the maps were predictive and of sufficient quality to compare the predicted and experimental structure of binding protein C8.

### CryoEM imaging of ChuA-G7 and ChuA-H3 complexes

ChuA (72.3 μM final concentration; 5 mg/ml) was mixed with binder G7 or H3 (72.3 μM final concentration) 2 h before grid preparation and 3.5 μl of the complex was applied onto a glow-discharged UltrAuFoil grid (Quantifoil GmbH) and were flash frozen in liquid ethane using the Vitrobot mark IV (ThermoFisher Scientific) set at 95% humidity and 4 °C for the prep chamber. Data were collected on a G1 Titan Krios microscope (ThermoFisher Scientific) with S-FEG as electron source operated at an accelerating voltage of 300 kV. A C2 aperture of 50 μm was used and no objective aperture was used. Data was collected at a nominal magnification of 105 K in nanoprobe EFTEM mode. Gatan K3 direct electron detector positioned post a Gatan BioQuantum energy filter was operated in a zero-energy-loss mode using a slit width of 10 eV to acquire dose fractionated images of the ChuA-G7 and ChuA-H3 complexes. One dataset was collected, composed of ~7000 movies. Movies were recorded in hardware-binned mode yielding a physical pixel size of 0.82 Å pixel$^{-1}$ with a dose rate of 8.5 e− pixel$^{-1}$ s$^{-1}$. An exposure time of 7 s yielded a total dose of 70.0 e− Å$^{-2}$, which was further fractionated into 70 subframes. Automated data collection was performed using EPU (ThermoFisher Scientific) with periodic centring of zero loss peak. A defocus range was set between −1.4 and −0.5 μm.

### CryoEM data processing and analysis

Micrographs from all datasets were motion-corrected using Motion-Cor 3.0 (Chan Zuckerberg Institute) and dose-weighted averages had their CTF parameters estimated using CTFFIND 4.1.8, implemented using Relion 5.0[59]. Particle coordinates were determined by crYOLO 1.7.6 using a general model[60]. 4x binned particles were extracted from micrographs using Relion 5.0, before being imported into cryoSPARC 4.4.1 for initial 2D classification to remove bad particles, followed by ab initio model generation and 3D refinement[61,62]. Subsequently, the data from each dataset was processed as described in Supplementary Fig. 3b, c with a final resolution of 2.97 Å for ChuA-G7 and 2.51 Å for ChuA-H3, respectively (FSC = 0.143, gold standard). An AlphaFold2 model of the ChuA-G7 and ChuA-H3 complexes were fitted into the CryoEM density maps before model improvement and refinement using PHENIX[55] and Coot[58].

### Biolayer interferometry analysis

BLI experiments were conducted using the Gator Plus Label-Free Bioanalysis System (Gator Bio). Ni-NTA probes were hydrated in binding buffer (50 mM Tris, 200 mM NaCl, 0.03% DDM, pH 7.8) for at least 10 minutes before use, and all experiments were performed at 25 °C with an orbital shaking speed of 1000 rpm. Samples were diluted in binding buffer and probes were then loaded with either binder, αβHb dimer or myoglobin to a shift of 0.2–0.4 nm, to avoid crowding on the probe surface. After loading probes were dipped in

buffer and then subsequently exposed to a 2-fold increasing ChuA concentration series, starting at 10 nM, to determine association signals. Between concentrations, probes were also dipped in binding buffer to determine dissociation signals. Each association-dissociation step lasted for five minutes to ensure enough binding and dissociation occurred before starting the next cycle. A loaded reference probe dipped in buffer was included as a control and was subtracted from sample probes during data analysis. For specificity screening of binders, binder-loaded probes were dipped in buffer and then either 500 nM FhuE, PqqL, YddB, or ChuA as a control, before being dipped in buffer. Analysis was conducted using the integrated Gator® GatorOne software (Gator Bio). Data was reference-subtracted, inter-step corrected (to association) and Savitzky-Golay filtered (for experiments with strong binding signal). Binding data was calculated using a combined association-dissociation 1:1 model, with global fitting (association window: 0–300 sec, dissociation window: 0–90 sec). Each experiment was performed in triplicate, and the $K_D$ values were averaged. Data was plotted using GraphPad Prism 10 Software.

### Statistical analysis

Data analysis was conducted using GraphPad Prism 10 software. Data are shown as mean ± s.e.m. or SD. For three or more groups statistical significance was determined by a one-way ANOVA with multiple comparisons tests. $P < 0.05$ was considered statistically significant. $IC_{50}$ and $EC_{50}$ values were calculated using a nonlinear regression (curve fit). BLI binding statistics were determined using the integrated Gator® GatorOne software (Gator Bio). Only full association and dissociation data with $R^2$ values > 0.98 and $X^2 < 3$ were included for analysis.

### Reporting summary

Further information on research design is available in the Nature Portfolio Reporting Summary linked to this article.

## Data availability

CryoEM maps and atomic models generated from this study have been deposited in the Protein Data Bank (PDB IDs: ChuA+Heme = 9DHE, binder C8 = 9DIV, ChuA+G7 = 9DIR, ChuA+H3 = 9DIS) and the Electron Microscopy Data Bank (EMDB IDs: ChuA+G7 = EMD-46916, ChuA +H3 = EMD-46917). All other data required to support the findings of the study are provided with this article. Source data are provided with this paper.

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

## Acknowledgements

The authors acknowledge the use of electron microscopy and cryo-sample preparation facilities at the Ian Holmes Imaging Centre of the Bio21 Molecular Science & Biotechnology Institute, the University of Melbourne; in particular, Dr. Hamish Brown for training and assistance, and at the Ramaciotti Centre for Cryo-Electron Microscopy of the Biomedicine Discovery Institute, Monash University. This research was supported by ARC LIEF grants (LE200100045, LE120100090) for the Titan Krios Gatan K3 Camera and the Titan Krios, and an ARC discovery project grant awarded to R.G. and G.K. (DP230102150). This research was undertaken on the MX2 beamlines at the Australian Synchrotron, part of ANSTO (CAP20894). R.G. and D.R.F. are members of the Australian Research Council Industrial Transformation Training Centre for Cryo-Electron Microscopy of Membrane Proteins for Drug Discovery (IC200100052). R.G. was funded by an NHMRC EL1 investigator grant (APP1197376). G.K. was supported by the Snow Medical Research Foundation (SMRF2021-276). D.R.F. was supported by an Australian Government Research Training Program (RTP) Scholarship. The authors thank Rebecca S Bamert for her contribution to the development of the small-scale expression and purification workflow used in this study. The authors thank Dr. Roxanne Smith at the Bio21-WEHI Crystallisation Facility, at the University of Melbourne, for her assistance with sample characterisation, crystallographic screening, and optimisation. The authors also acknowledge the use of the facilities at the Melbourne Protein Characterisation Facility, of the Bio21 Molecular Science & Biotechnology Institute, the University of Melbourne and thank Belinda Michell and Troy Attard for training and assistance. The authors also acknowledge the WEHI Protein Production Facility. The authors thank Dr. Vicki Sifniotis from Solve Scientific Pty Ltd for help with Gator BLI analysis. The authors thank Dr. Geoffrey Kong at the Monash Macromolecular Crystallisation Platform at Monash University for assistance with crystallisation screening and DSF experiments. The authors also thank Dr. Ben Adler at UC Berkeley for supply of the pTET plasmid used in this study.

## Author contributions

Conceived and designed the experiments: D.R.F., K.A., M.D., G.K., and R.G.; performed the experiments: D.R.F., K.A., I.S., B.A.S., A.K., C.J.L., K.L., H.V., M.D., and R.G., analysed the data: D.R.F., K.A., I.S., B.S., H.V., M.D., and R.G.; contributed reagents/materials/analysis tools: H.V., M.D., G.K., and R.G.; wrote and edited the manuscript: D.R.F. and R.G.; acquired funding and provided project supervision: R.G. and G.K. All authors edited and approved the manuscript.

## Competing interests

The authors declare no competing interests.

## Additional information

[1]Department of Microbiology, Biomedicine Discovery Institute, Monash University, Clayton, Australia. [2]Centre for Electron Microscopy of Membrane Proteins, Monash Institute of Pharmaceutical Sciences, Parkville, Victoria, Australia. [3]Department of Biochemistry and Pharmacology, Bio21 Molecular Science and Biotechnology Institute, The University of Melbourne, Parkville, Victoria 3010, Australia. [4]Department of Biochemistry and Molecular Biology, Biomedicine Discovery Institute, Monash University, Clayton 3800, Australia. [5]The Walter and Eliza Hall Institute of Medical Research, Parkville, Victoria 3052, Australia. [6]Ramaciotti Centre for Cryo-Electron Microscopy, Biomedicine Discovery Institute, Monash University, Clayton 3800, Australia. [7]Department of Medical Biology, University of Melbourne, Parkville, Victoria 3010, Australia. ✉e-mail: gavin.knott@monash.edu; rhys.grinter@unimelb.edu.au

