## [Transparent Peer Review file · Nature Communications]

Inhibiting heme-piracy by pathogenic *Escherichia coli* using de novo-designed proteins

Corresponding Author: Dr Rhys Grinter

Version 0:

Reviewer comments:

Reviewer #1

(Remarks to the Author)

It is clear that iron plays a critical role in most organisms and that the acquisition of iron, which typically has very low bioavailability, is of paramount importance. This is especially true for bacterial pathogens, given that the host often restricts access to iron (nutritional immunity). To obtain iron, bacteria have developed iron-acquisition strategies such as the secretion of siderophores, small molecules that bind iron with very high affinities and which are subsequently imported across the OM (in Gram-negative bacteria) by TBDTs, which are active transporters that utilise the proton motive force and the ExbBD-TonB complex in the inner membrane for import. Another iron acquisition strategy is the extraction (again by TBDTs) of (iron-containing) heme from heme-binding proteins such as Hemoglobin (Hb) and Myoglobin (Mb), and the subsequent import of the liberated heme. One example of such a TBDT is ChuA.

The manuscript by Fox et al. describes the detailed structural and functional characterisation of the ChuA iron acquisition TonB dependent transporter (TBDT) of pathogenic *E. coli* and *Shigella* strains, and design de novo binders that interfere rather strikingly with ChuA function. Overall, the goal was to uncover the structural basis for hemoglobin binding, heme extraction and import, via structural modelling and protein design, cryo-EM, X-ray crystallography, mutagenesis, and phenotypic analysis. Prior to this study it was already known that ChuA rapidly extracts heme from Hb and that His420 and His86 are important for this process.

The authors start off by generation of an *E. coli* strain lacking all TBDTs and complemented with the *chu* operon (ChuOP) on the genome. The *chuA* gene is deleted from this strain and can then be complemented by *chuA* on a plasmid. They show that the strains lacking ChuA grow on Hb, Mb and hemin, but not on unrelated iron-binding proteins such as neuroglobin and ferredoxin, showing that the heme extraction is specific. Hb binds with high affinity (~50 nM) as inferred from growth curves and corresponding EC50 values (Fig. 1). They then present an xX-ray crystal structure of the ChuA-hemin complex. Given that the input consisted of ChuA and Hb, the structure suggest (and confirms previous data) that the ChuA-Hb complex is transient and Hb is extracted rapidly. Hb is bound in the structure to His420, again confirming previous data. The same results were obtained via cryo-EM. Modelling with AF generated plausible models where the Hb (and Mb) interacts with loops L7 and L8 of ChuA. The heme binding site within Hb is close (8.5 Å for the iron) to the observed site in the crystal structure. Significantly, overlay of the model and structure suggest clashes by several side chains, providing suggestions for heme dislocation from its Hb binding site by ChuA loop residues. The AF models were supported by previously determined crystal structures of Hb binders (e.g. haptoglobin) as well as plate growth assays using Hb and Hb-complexes in various concentrations. Site directed mutagenesis is then used to confirm the involvement of His420 and His86 in heme binding. Interestingly, the data suggest that both these residues can bind to heme and are in accordance with AF predictions that suggest the same. Figure 4 then describes the generation of de novo ChuA-specific binders (to the L7 and L8 loops that are also involved in Hb/Mb binding), show that these inhibit heme extraction, and characterise the affinities to ChuA via growth assays and BLI. This is very nice data. Finally, in Fig. 5 the authors present nice cryoEM structures of ChuA complexed to two of the most effective binders, which convincingly show that these, via partial overlap with the Hb/Mb binding site, will indeed interfere with heme extraction, but not with heme import in accordance with their data showing that heme import still occurs in the presence of the binders.

This is a very nice and well-written paper that was a pleasure to read. My co-reviewer and I don't have any major concerns regarding the data or the conclusions drawn. The figures are also well-made. The design and production of specific ChuA binders that abolish its normal function of iron extraction from Hb is particularly interesting. To my knowledge, this kind of

approach, where binders are designed to block OMP-mediated uptake of essential nutrients, has not previously been reported for any OMP. I'm not sure whether protein-based binders will make the best future drugs, but the principle is really interesting. We only have some minor issues and requested areas of clarification, listed below.

1. Page 3: the paragraph describing the various pathogenic E. coli strains etc is perhaps a bit too long.
2. Line 15 (main): "may not be the transporter's preferred substrate". I think I know what the authors mean but this could be rephrased, since free heme/hemin is the transported substrate.
3. Lines 20-22 (main): Authors could state that the wt strain BW25113 does not have the Chu operon. BTW it is the preference of these reviewers to have the line numbers NOT start at 1 for very page.
4. Was BLI done for Heme or αHb but not shown or was it not done at all? The result for heme binding might be particularly interesting. But perhaps hemin is too small for BLI or there might be solubility issues?
5. Figure 2 m labels are upside-down. It would also be useful to state in the legend that the plates show 2-fold dilutions of each binder. Also applies to other figures where such data are shown. Figure 2 n, although easy to understand, it would help to explain in the legend that the values are a ratio compared to αHb . In Figure 2 the Mb colour is too similar to heme. Could the authors reconsider the colour of the protein? In Figure 2 e, h and f: the word hemin is difficult to see.
6. Have the authors tried to predict the structure of the mutant ChuA1423A;N429A;F484A and compare it to that of the WT? And have they tried to predict this structure using AlphaFold multimer to see if the ChuA- αHb interface is still maintained despite the mutations?
7. Figure 3 c: label upside down, difficult to read.
8. Page 12, line 24. Considering ChuA a "flexible membrane protein target" is perhaps a bit overstated. We agree that the L7 and L8 loops might be dynamic to some extent but overall ChuA appears to be a well-behaved OMP.
9. When the authors state: "We generated ~20,000 ChuA binders in silico and selected 96 designs for wet lab screening using AlphaFold2 filtering and manual curation" what was the rationale for the manual curation? The manuscript will benefit if a better explanation was given about the process of generating the binders and the selection parameters chosen, especially given that this is a novel approach that will be of interest to many. So please try to give as much detail as possible in this section.
10. page 13 line 8: please add that the pET20 vector has a PelB signal sequence for periplasmic expression.
11. Protein production: please add typical yields obtained for the various proteins purified. Also related to protein purification: how were the globins loaded with heme?
12. Page 16, line 15: sentence is garbled.
13. Page 20: crystallisation conditions; correct "Acet, Tart".
14. Data availability: please add which ID corresponds to which structure and map.
15. Fig. 1: what is the significance of the EC50 label colours?
16. Fig. S1a: i don't find the difference particularly convincing since the delta chuA strain also appears to grow. Maybe too many cells were streaked out?
17. Fig. S2: did the authors try to solve the ChuA-hemin structure via cryoEM?
18. Fig. S5e,f: it would be useful to have an Hb panel here as well to see inhibition (positive control). Legend line 9: de novo binders. Legend line 12: small letters k for koff and kon.
19. Supplemental Tables: separating the table titles and actual tables is not helpful, please combine them.
20. Table S2: Both structures have clash scores a bit higher than optimal values, can the authors try to improve these parameters? Rwork and Rfree from Binder C8 are very close suggesting overfitting to the data. Can you add the molprobit scores for both structures?

Reviewer #2

(Remarks to the Author)

Reviewer #3

(Remarks to the Author)

The authors present a comprehensive and multifaceted study that directly addresses the a crucial mechanism for iron acquisition from heme by a prevalent enteric pathogen. Specifically, how the outer membrane heme transporter ChuA functions in binding the alpha-beta form of hemoglobin. The work also resolves a long standing hypothesis, observed during the investigation of a number of outer membrane heme transporters from gram negative pathogens. Specifically, that a number of amino acids may be involved in heme ligation during transport and that they can often substitute for one another. While the authors do not test their hypotheses under conditions relevant to a chronic infection (anaerobic gut model etc.), there are two impressive aspects of this work are; 1) the fact that the authors look at in-vivo heme survival using the Chu operon and a number of variants and 2) use the structural knowledge gained in the work to generate de novo peptides that will disrupt specific protein-protein interactions required for the liberation of heme from the alpha-beta hemoglobin by ChuA. This reviewer found the interference by hemopexin somewhat comforting and an additionally important finding that may warrant further exploration. One wonders what the binders would do in a similar assay (Figure 2, panel m and n), and why that experiment wasn't done. One other minor suggestion; The colors used in Figure 5 are very similar in pigmentation and this color-blind reviewer has a hard time interpreting the demarcations. It may be worth remaking the figure using opposing colors on the RGB wheel to really make the interfaces pop.

Reviewer #4

(Remarks to the Author)

In this work, Fox et al present the structure of the ChuA protein from E coli and provide insights into its function as a heme transporter by clarifying the role of two important amino acid residues, His-86 and His-420, in importing hemin and heme. The authors then use these structures along with their functional insights to generate de novo designed binding proteins and characterize their best designs in vitro by inhibiting E coli growth. Noteworthy results include that this appears to be the first experimentally determined structure of ChuA from E coli, and to my knowledge, this is the first de novo protein designed to bind ChuA. While de novo design of protein binders to target structures is not particularly novel at this stage, the data provided show that proteins targeting regions of the ChuA protein could be a viable and novel strategy for antimicrobial treatment. I can comment on the structural studies as well as the de novo protein design approaches but interpretation of the importance of these findings in the context of the field are best left to other reviewers.

Overall, the work may be of interest to members to this specific community, and more structural data is always great; indeed, the structural work presented in the manuscript is well done. But my interpretation is that the protein design work done here is not particularly groundbreaking and that the experimental results shown in Figure 4 are sufficient to show that these proteins work as designed, but motivate deeper questions about more therapeutically relevant assays. While it might go beyond the scope of this manuscript as written, experimental assays demonstrating the antimicrobial properties of these proteins in a therapeutic context would greatly elevate this study. Other studies presenting structural biology and protein design previously published in Nature Communications (for example, Roy et al 2023 <https://doi.org/10.1038/s41467-023-41272-z> and Lv et al 2024 <https://doi.org/10.1038/s41467-024-52582-1>) have included more extensive in vitro or in vivo data and I would prefer to see something comparable in this study for publication with this journal. The methods and design approaches, though, are sound.

Specific points:

The crystal structure of ShuA from *Shigella dysenteriae* (PDB ID 3FHH) was reported more than a decade ago and is highly similar to this structure (99% amino acid identity, RMSD = 0.430 Å). Discussion of the importance of this ChuA structure relative to the previously reported ShuA structure would be useful in understanding the impact of these new experimentally determined structures.

General discussion of the potential cross-reactivity of these binders would be useful for interpreting these findings, especially because the authors present these designed proteins as potential antimicrobials with (I would assume) therapeutic hopes. In the introduction they discuss who these are important targets for pathogenic bacteria, but also note that "most bacteria require iron as a cofactor" (line 24) - would this include bacteria in the human microbiome? Designed proteins specific only to pathogenic strains would be of greater interest.

Similarly, can authors comment on the potential of the de novo proteins being cross-reactive towards ShuA? Or was any modeling work done to predict the structures of these de novo binders with ShuA? With such similar amino acid sequences and overall structures, these de novo binders could easily be cross-reactive, and that would increase the impact of these findings. If that is not the case, I would be curious to hear why.

In the "Design of de novo binders" methods section, some insight into what "manually curated" means would be useful.

Figure 1b - Labels for the plates (such as deltaTBDT) would be more readable if they were oriented in the same way as ChuOP. The size of the axis label and tick marks on the accompanying graphs should also be made larger; they are difficult to read.

Figure 2m - the concentration label on the abHb-Hp plate is upside down (I believe)

Figure 3 a-h - similar remarks on concentration labeling. Personally I would like the structural images to be larger in the insets in c-h to better highlight the structural findings in this paper, which I believe are a strength of the manuscript.

Figure S4 - the computational models are so similar to the final structure. I think this would be a good opportunity to discuss why finding the structure experimentally was important; one interpretation of this image could be "why solve this structure if the model is so accurate".

Version 1:

Reviewer comments:

Reviewer #1

(Remarks to the Author)

The authors have addressed our comments and concerns in a satisfactory manner. One issue remains, related to point 11: can the authors please add how the globins were loaded with heme?

Reviewer #2

(Remarks to the Author)

Reviewer #4

(Remarks to the Author)

The authors have addressed the points I raised in my initial review.

RESPONSE TO REVIEWER COMMENTS

We thank the reviewers for the time spent reviewing our manuscript and the insightful and constructive suggestions. Responses to the specific comments are outlined below.

Reviewer #1 (Remarks to the Author):

1. Page 3: the paragraph describing the various pathogenic E. coli strains etc is perhaps a bit too long.

We have significantly shortened this section in the revised manuscript.

2. Line 15 (main): "may not be the transporter's preferred substrate". I think I know what the authors mean but this could be rephrased, since free heme/hemin is the transported substrate.

We agree with the reviewer that this statement is ambiguous, we have revised it to 'may not be the transporter's preferred target for extracellular uptake' (Revised manuscript, line 131)

3. Lines 20-22 (main): Authors could state that the wt strain BW25113 does not have the Chu operon. BTW it is the preference of these reviewers to have the line numbers NOT start at 1 for very page.

This is a good suggestion we have modified this section of the manuscript so that it references the fact that BW25113 does not encode the Chu operon (Revised manuscript Lines 136-140)

To solve this, we utilised an E. coli BW25113 strain that lacks all TBDTs involved in iron uptake (E. coli Δ TBTD)³⁰⁻³². E. coli BW25113 does not naturally possess the Chu operon, which encodes ChuA and other proteins required heme import, so we inserted this operon by homologous recombination^{16,34} (Figure 1a).

We have changed to continuous line numbering in the revised manuscript

4. Was BLI done for Heme or ayHb but not shown or was it not done at all? The result for heme binding might be particularly interesting. But perhaps hemin is too small for BLI or there might be solubility issues?

We were unsuccessful in our attempts to measure the heme binding to ChuA by BLI. Heme solubility isn't an issue. However, when ChuA was immobilised on the BLI surface we didn't detect any signal for heme binding. We also didn't detect any signal for binding of our de novo binding proteins in this configuration, suggesting that the change in conformation/mass of the resulting complex was not enough to

generate a measurable signal. This experiment is likely further complicated by the fact that ChuA is in a lipid micelle, which increases the size of the molecule and may mask signals when associated with BLI surface binding. To obtain our binder-ChuA association/disassociation curves, we immobilised the binders on the BLI surface (via a 6xhis-tag) and measured the association with free ChuA. This approach isn't practical for hemin as it would need to be immobilised on the BLI surface by chemical modification, which due to its size would make it inaccessible for ChuA binding due to steric hindrance.

We didn't analyse the α Hb-ChuA interaction via BLI. Considering the similar SC50 for the two proteins we didn't feel this experiment would provide much additional insight. The production of ChuA is resource intensive, and we didn't think the experiment was justified considering it would require a significant amount of the transporter to perform and optimise.

5. Figure 2 m labels are upside-down. It would also be useful to state in the legend that the plates show 2-fold dilutions of each binder. Also applies to other figures where such data are shown. Figure 2 n, although easy to understand, it would help to explain in the legend that the values are a ratio compared to $\alpha\beta$ Hb. In Figure 2 the Mb colour is too similar to heme. Could the authors reconsider the colour of the protein? In Figure 2 e, h and f: the word hemin is difficult to see.

We have addressed these issues with Figure 2 in the revised manuscript.

6. Have the authors tried to predict the structure of the mutant ChuA1423A;N429A;F484A and compare it to that of the WT? And have they tried to predict this structure using AlphaFold multimer to see if the ChuA- $\alpha\beta$ Hb interface is still maintained despite the mutations?

AlphaFold2/3 and related deep learning-based protein structure prediction methods aren't well suited to determining the effects of point mutations on a protein structure. This is because these methods train a neural network on protein structure and sequence patterns, as opposed to calculating physical forces, and their training dataset does not contain any information on the effects of these mutations on the protein. This means that AlphaFold's prediction of a mutated protein usually does not take account the effect of the mutation and produces a model structurally analogous to the wildtype protein. Consistent with this, for ChuA1423A;N429A;F484A AlphaFold2/3 produces a model very similar to the wildtype ChuA- $\alpha\beta$ Hb complex, which we elected not to present because it may lack physiological relevance. In contrast to this, we were comfortable presenting the AlphaFold3-heme binding data for the ChuA histidine mutants in Figure 4d of the revised manuscript, as these histidines are individually critical for heme coordination, so we feel these models are much more likely to be accurate.

7. Figure 3 c: label upside down, difficult to read.

This has been corrected in the revised manuscript

8. Page 12, line 24. Considering ChuA a "flexible membrane protein target" is perhaps a bit overstated. We agree that the L7 and L8 loops might be dynamic to some extent but overall ChuA appears to be a well-behaved OMP.

We agree that this section did not quite capture the essence of ChuA as a binder target. We have revised this section of the manuscript to improve this (Revised manuscript, lines 365-370):

'Two of these binders achieved sub-100 nM affinity, without experimental optimisation, validating AI-guided protein design for binder generation against flexible extracellular regions of integral membrane protein targets.'

9. When the authors state: "We generated ~20,000 ChuA binders in silico and selected 96 designs for wet lab screening using AlphaFold2 filtering and manual curation" what was the rationale for the manual curation? The manuscript will benefit if a better explanation was given about the process of generating the binders and the selection parameters chosen, especially given that this is a novel approach that will be of interest to many. So please try to give as much detail as possible in this section.

For binder generation, we followed the method outlined in the research papers and associated GitHub repositories for the deep learning protein design tools we utilised in this study, RFdiffusion¹ (<https://github.com/RosettaCommons/RFdiffusion>) and DL binder design² (https://github.com/nrbennet/dl_binder_design), which we referenced in the manuscript. The creators of these packages have done a really good job of making them accessible to non-expert users, so we didn't feel like we needed to go into too much detail.

We provided a description of our binder design process in the materials and methods. In the revised manuscript we have expanded the description of our filtering and selection process (Revised manuscript, lines 566-571).

'These models were manually curated to maximise structural variability (length, helix topology, α and β secondary structure), to maximise binder-ChuA interaction interface area, and based on suitability for affinity tagging, to identify 96 designs for synthesis and testing.'

10. page 13 line 8: please add that the pET20 vector has a PelB signal sequence for periplasmic expression.

We have added this information to the revised manuscript (Revised manuscript, line 382).

11. Protein production: please add typical yields obtained for the various proteins purified. Also related to protein purification: how were the globins loaded with heme?

We have added average yields for the protein purifications performed in the manuscript.

12. Page 16, line 15: sentence is garbled.

This has been corrected in the revised manuscript (Revised manuscript, lines 488-489).

13. Page 20: crystallisation conditions; correct "Acet, Tart".

This has been corrected in the revised manuscript.

14. Data availability: please add which ID corresponds to which structure and map.

This has been added to the revised manuscript.

15. Fig. 1: what is the significance of the EC50 label colours?

These colours represent the potency of the growth stimulatory effect. We have added a description of this to the figure legend of the revised manuscript

16. Fig. S1a: i don't find the difference particularly convincing since the delta chuA strain also appears to grow. Maybe too many cells were streaked out?

This experiment has been repeated for the revised manuscript and the growth difference is more convincing now. The initial plate was overgrown due to heavy streaking leading to the carry-over of iron from the starter culture.

17. Fig. S2: did the authors try to solve the ChuA-hemin structure via cryoEM?

Considering we already have a crystal structure and credible AlphaFold models for this complex we did not attempt to determine the structure by cryoEM.

18. Fig. S5e,f: it would be useful to have an Hb panel here as well to see inhibition (positive control).

Panels showing binder inhibition when grown on Hb and Mb are shown in Figure 4b,c. It's relatively easy for the reader to cross reference this figure with Figure S5,

and we would rather not present duplicate data in the paper, so we have not implemented this suggestion.

Legend line 9: de novo binders. Legend line 12: small letters k for koff and kon.

Corrected in the revised manuscript

19. Supplemental Tables: separating the table titles and actual tables is not helpful, please combine them.

This occurred because of how the tables are handled by the Nature Communications submission portal. This will be corrected in the final version of the manuscript.

20. Table S2: Both structures have clash scores a bit higher than optimal values, can the authors try to improve these parameters? Rwork and Rfree from Binder C8 are very close suggesting overfitting to the data. Can you add the molprobity scores for both structures?

We put considerable effort into modelling these structures. However, the quality of the experimental data and maps isn't the best (relatively low resolution, some anisotropy), so it was difficult to obtain the best clash statistics for these models. However, the clash scores are quite reasonable compared to structures of the same resolution range (95th percentile for structure >2.95 Å for ChuA+Heme; 89th percentile for structures 2.46 Å +/- 0.25 Å for the binder C8); The MolProbity score for these structures is also illustrative of their quality given the resolution of the data (1.19, which is in the 100th percentile 3.2 Å +/- 0.25 Å for ChuA+Heme; and 1.94 which is in the 96th percentile 2.46 Å +/- 0.25 Å for binder C8). The MolProbity scores for these structures have been included in the structure data table in the revised manuscript.

Overfitting of a crystallographic model to the data is reflected by a large difference (typically larger than 7-10%) between the Rwork and Rfree values. The gap between these values in the C8 binder structure is only 2.2% indicating that the model is not overfitted. As noted in the methods section, the C8 crystal we analysed suffered from strong translational non-crystallographic symmetry (tNCS). In addition to making the structure challenging to solve by molecular replacement (luckily, we had a near-perfect molecular replacement model), tNCS leads to inflated R-values, which is reflected in the higher values for Rwork and Rfree in this structure. Despite this, the maps remained robustly interpretable and allowed us to accurately model the eight C8 binder molecules in the asymmetric unit.

Reviewer #2 (Remarks to the Author):

Comments included with those of reviewer 1

Reviewer #3 (Remarks to the Author):

This reviewer found the interference by hemopexin somewhat comforting and an additionally important finding that may warrant further exploration. One wonders what the binders would do in a similar assay (Figure 2, panel m and n), and why that experiment wasn't done.

We attempted this experiment for the revisions, with the ChuA binder added to the agar plate and Hb or Hb-Hp dilutions spotted as in Figure 2 m/n. However, we had difficulty titrating the amount of binder so that we saw growth from Hb and Hb-Hp but saw a consistent synergistic effect from Hp and the binder. We only had a limited amount of haptoglobin and so were unable to rigorously optimise this experiment. While the proposed experiment is interesting, we don't think it is critical for any of the conclusions of our current manuscript, so we elected not to include it in the revised manuscript.

One other minor suggestion;

The colors used in Figure 5 are very similar in pigmentation and this color-blind reviewer has a hard time interpreting the demarcations. It may be worth remaking the figure using opposing colors on the RGB wheel to really make the interfaces pop.

We appreciate the reviewer's suggestions for making our manuscript more accessible. We have changed the colour for binders G7 (to deep orange) and H3 (to a deeper blue) in this figure and elsewhere in the manuscript. We have checked the resulting figures by conversion to black and white, and in an online colour blindness visualiser and feel that the new colours clearly differentiate the different structural components on display under these conditions.

Reviewer #4 (Remarks to the Author):

Overall, the work may be of interest to members to this specific community, and more structural data is always great; indeed, the structural work presented in the manuscript is well done. But my interpretation is that the protein design work done here is not particularly groundbreaking and that the experimental results shown in Figure 4 are sufficient to show that these proteins work as designed, but motivate deeper questions about more therapeutically relevant assays.

We appreciate the time and effort the reviewer took to review our manuscript. While the protein design work in the manuscript applied existing tools (RFdiffusion, ProteinMPNN, AlphaFold2) to generate our ChuA binders, we feel that our results are groundbreaking for several reasons. This study is one of few studies to apply to apply deep learning-based de novo protein design to design binders with biological function. Further, it is one of the first studies using these tools that isn't authored by the laboratories that are actively developing this software (e.g. the Baker lab). So,

this paper shows that these tools can be used successfully by the wider community. To our knowledge, this is the first study where de novo-designed binders have been generated for an integral outer membrane protein, a target class that has several additional difficulties. As noted by reviewer 1, this is the first study to de novo design protein inhibitors of a bacterial outer membrane protein, and the first to show that de novo-designed proteins can be applied to inhibit bacterial growth by blocking nutrient import across the bacterial outer membrane.

We respectfully disagree with the reviewer that the data in Figure 4 (Figure 5 in the revised manuscript) is sufficient to show the proteins are working as designed, from a structural perspective at least. It was exciting and gratifying that our experimental structures of the ChuA-binder complexes matched the computational designs so closely, and we feel this is a significant finding. We do not think our results would be so convincing to the research community without these experimental structures. The deep learning approach we applied to generate binders is very new (the initial preprint and code were released in December 2022), and while some other papers and preprints show the computational designs closely match experimental structures, it was important for us to solve the experimental structures of our complexes, to convince ourselves and others of the fidelity of this fully de novo design process. By going to the effort of producing these structures we validate the technique for others interested in using it and decrease their need for performing experimental structural validation.

While it might go beyond the scope of this manuscript as written, experimental assays demonstrating the antimicrobial properties of these proteins in a therapeutic context would greatly elevate this study. Other studies presenting structural biology and protein design previously published in Nature Communications (for example, Roy et al 2023 <https://doi.org/10.1038/s41467-023-41272-z> and Lv et al 2024 <https://doi.org/10.1038/s41467-024-52582-1>) have included more extensive in vitro or in vivo data and I would prefer to see something comparable in this study for publication with this journal.

While our development of de novo-designed protein ChuA inhibitors is exciting, it is far from the only novel aspect of our manuscript. We present a comprehensive body of experimental work that sheds light on the mechanism that ChuA uses to obtain heme from host hemoglobin. These findings inform and complement the binder design aspect of the study and together these data represent a significant advance in our understanding, in the absence of preclinical work aimed at the therapeutic application of the ChuA binders. While the development of these protein binders as therapeutics is something we intend to pursue it is outside the scope of this current study.

The two papers referenced by the reviewer do not significantly advance our understanding of the biology of the target of the protein binders but rather focus on binder design and testing. This makes the scope of these studies distinct from ours and makes the inclusion of preclinical data in their studies appropriate. Additionally, the methods employed by these papers to generate protein binders are distinct from the RFdiffusion-ProteinMPNN pipeline we use in our work. The study on integrin inhibitors uses an experimental structure of a bound peptide as a starting point for

protein design, and while this study is impressive, it is debatable whether this approach is truly de novo. Alternatively, the study targeting C. difficile toxin B, used a screening approach to experimentally test many 1000s of designs to identify binders with high affinity to the target. This is in comparison to the 96 computational designs we experimentally tested in our study. The limited number of designs we tested is exciting and important because it makes protein design much more accessible to labs without lab the funds and infrastructure for large-scale gene synthesis and automated binder screening.

The methods and design approaches, though, are sound.

Specific points:

The crystal structure of ShuA from *Shigella dysenteriae* (PDB ID 3FHH) was reported more than a decade ago and is highly similar to this structure (99% amino acid identity, RMSD = 0.430 Å). Discussion of the importance of this ChuA structure relative to the previously reported ShuA structure would be useful in understanding the impact of these new experimentally determined structures.

ShuA and ChuA are essentially the same protein. The proteins only differ by two conservative amino acid substitutions (V61I, and D234E) that do not form part of the heme/hemoglobin binding interface. As such, in the manuscript, we introduced the transporter as ChuA (aka. ShuA). To make this point clearer in the revised manuscript we have modified the sentence comparing our ChuA structure with the previously solved structure of ShuA (Lines 174-175) so that it reads:

Aside from bound heme, the overall structure was highly similar to the previously solved apo-structure of the essentially identical transporter ShuA¹⁸ (99% AA identity; RMSD = 0.430 Å out of 3626/4638 atoms).

General discussion of the potential cross-reactivity of these binders would be useful for interpreting these findings, especially because the authors present these designed proteins as potential antimicrobials with (I would assume) therapeutic hopes. In the introduction they discuss who these are important targets for pathogenic bacteria, but also note that “most bacteria require iron as a cofactor” (line 24) - would this include bacteria in the human microbiome? Designed proteins specific only to pathogenic strains would be of greater interest.

*Due to the specific nature of the protein-protein interactions between ChuA and our binders (and de novo-designed binders in general), we do not feel that high-affinity cross-binding with other proteins is likely. To test this, in the revised manuscript we performed additional BLI experiments to assess the interaction between our three top ChuA binders (A10, G7, and H3), and two other *E. coli* TBDTs (YddB and FhuE), and an *E. coli* periplasmic protease (PqqL). In contrast to ChuA, which served as a positive control for these experiments, no binding signal was observed for any of these proteins with any of the binders. We have included these data in Figure S5i and discussed it in the revised manuscript (Lines 310-314). Based on these experiments, we do not expect that the ChuA binders will bind to other TonB-*

dependent transporters involved in heme or iron uptake, any more than they would bind to the control proteins we tested.

Our binders will likely inhibit heme extraction from hemoglobin by E. coli strains and other (entero)bacteria that produce a close homologue of ChuA. Considering the major physiological substrate of ChuA is hemoglobin, we expect our binders will be most relevant in an infection context, where this substrate is available. In agreement with this, ChuA is a virulence factor that is generally encoded by pathogenic E. coli³. As such, we feel it is likely these binders will be specific to pathogenic strains of E. coli.

Similarly, can authors comment on the potential of the de novo proteins being cross-reactive towards ShuA? Or was any modelling work done to predict the structures of these de novo binders with ShuA? With such similar amino acid sequences and overall structures, these de novo binders could easily be cross-reactive, and that would increase the impact of these findings. If that is not the case, I would be curious to hear why.

As discussed above, ShuA is identical to ChuA in the heme/hemoglobin binding region, and as such our binders would definitely also bind to and inhibit this protein. It is for this reason we discuss Shigella briefly in our introduction. These binders may also have utility in treating shigella infections. However, because we aren't focusing directly on preclinical validation or the therapeutic applications of our binders in this paper, we would prefer not to discuss this at this stage.

In the "Design of de novo binders" methods section, some insight into what "manually curated" means would be useful.

As discussed in our response to reviewer 1, we have expanded our description of how we performed manual curation of our binders in the revised manuscript (Revised manuscript, lines 566-569).

'These models were manually curated to maximise structural variability (length, helix topology, α and β secondary structure), to maximise binder-ChuA interaction interface area, and based on suitability for affinity tagging, to identify 96 designs for synthesis and testing.'

Figure 1b - Labels for the plates (such as deltaTBDT) would be more readable if they were oriented in the same way as ChuOP. The size of the axis label and tick marks on the accompanying graphs should also be made larger; they are difficult to read.

These suggested changes have been made to the figure in the revised manuscript

Figure 2m - the concentration label on the abHb-Hp plate is upside down (I believe)

This has been corrected in the revised manuscript.

Figure 3 a-h – similar remarks on concentration labelling. Personally, I would like the structural images to be larger in the insets in c-h to better highlight the structural findings in this paper, which I believe are a strength of the manuscript.

We agree with the reviewer that these structural panels were too small to be viewed effectively. This was largely due to the amount of data present in Figure 3. To remedy this, we have rearranged the figure panels in Figure 3 and split the figure into two figures (Figures 3 and 4 in the revised manuscript).

Figure S4 – the computational models are so similar to the final structure. I think this would be a good opportunity to discuss why finding the structure experimentally was important; one interpretation of this image could be “why solve this structure if the model is so accurate”.

The fully de novo protein design methods we used in this paper are very new. The designs do not start from an existing protein fold or scaffold. The binder folds are generated by a diffusion network, and the sequence is assigned using a neural network, generating proteins that are completely new to nature. While the authors of the original paper describing this method¹ and a few subsequent papers/prints present experimental data showing strong agreement between the AlphaFold prediction of their binder complexes and experimental structures, we thought it was important to solve the experimental structure of our binder complexes to convince the community of the accuracy of this method. Rather than discussing this in detail, we would prefer to let the results speak for themselves.

References

- 1 Watson, J. L. *et al.* De novo design of protein structure and function with RFDiffusion. *Nature* **620**, 1089-1100 (2023).
- 2 Bennett, N. R. *et al.* Improving de novo protein binder design with deep learning. *Nature Communications* **14**, 2625 (2023).
- 3 Hagan, E. C. & Mobley, H. L. Haem acquisition is facilitated by a novel receptor Hma and required by uropathogenic *Escherichia coli* for kidney infection. *Molecular microbiology* **71**, 79-91 (2009).